# Dysfunctions of the paraventricular hypothalamic nucleus induce hypersomnia in mice

**Chang-Rui Chen[1]\*[†], Yu-Heng Zhong[1†], Shan Jiang[1†], Wei Xu[1†], Lei Xiao[1], Zan Wang[2], Wei-Min Qu[1]\*, Zhi-Li Huang[1]\***

[1]Department of Pharmacology, School of Basic Medical Sciences; State Key Laboratory of Medical Neurobiology and MOE Frontiers Center for Brain Science, Institutes of Brain Science, Fudan University, Shanghai, China; [2]Department of Neurology, The First Hospital of Jilin University, Changchun, China

**\*For correspondence:**
changruichen@163.com (C-RC);
quweimin@fudan.edu.cn (W-MQ);
huangzl@fudan.edu.cn (Z-LH)

[†]These authors contributed equally to this work

**Competing interest:** The authors declare that no competing interests exist.

**Abstract** Hypersomnolence disorder (HD) is characterized by excessive sleep, which is a common sequela following stroke, infection, or tumorigenesis. HD is traditionally thought to be associated with lesions of wake-promoting nuclei. However, lesions of a single wake-promoting nucleus, or even two simultaneously, did not exert serious HD. Therefore, the specific nucleus and neural circuitry for HD remain unknown. Here, we observed that the paraventricular nucleus of the hypothalamus (PVH) exhibited higher c-fos expression during the active period (23:00) than during the inactive period (11:00) in mice. Therefore, we speculated that the PVH, in which most neurons are glutamatergic, may represent one of the key arousal-controlling centers. By using vesicular glutamate transporter 2 (vglut2$^{Cre}$) mice together with fiber photometry, multichannel electrophysiological recordings, and genetic approaches, we found that PVH$^{vglut2}$ neurons were most active during wakefulness. Chemogenetic activation of PVH$^{vglut2}$ neurons induced wakefulness for 9 hr, and photostimulation of PVH$^{vglut2}$→parabrachial complex/ventral lateral septum circuits immediately drove transitions from sleep to wakefulness. Moreover, lesioning or chemogenetic inhibition of PVH$^{vglut2}$ neurons dramatically decreased wakefulness. These results indicate that the PVH is critical for arousal promotion and maintenance.

## Editor's evaluation

Your work demonstrating a novel role of a projection from the paraventricular nucleus of the hypothalamus to the parabrachial nucleus in regulating wakefulness will be of interest to the readership of *eLife*. In particular, your study will add to the fields of sleep and hypothalamic research.

## Introduction

Hypersomnolence disorder (HD) is characterized by an irresistible need for sleep and an inability to stay awake during major waking episodes, which results in reduced function and overall worse life quality, highlighting its public health importance (*Mahowald and Schenck, 2005*). However, few dysfunctional wake-promoting nuclei have been identified to induce hypersomnia. Therefore, further identification of key hypersomnia-controlling nuclei and neural circuitry represents a common goal for clinicians and researchers.

In the last 100 years, more than 16 wake-promoting nuclei have been identified. Von Economo first proposed a hypersomnia-controlling region located in the posterior hypothalamus from observations of marked somnolence in patients with epidemic encephalitis lethargic (*Saper et al., 2001*).

Furthermore, Moruzzi et al. and other studies have revealed that a brainstem ascending reticular activating system (ARAS) is responsible for wakefulness (*Moruzzi and Magoun, 1949*; *Xu et al., 2015*; *Carter et al., 2010*). However, cell-body-specific ablation or inhibition of components of the ARAS—including the laterodorsal tegmentum (LDT), basal forebrain (BF), pedunculopontine tegmental nucleus (PPT) cholinergic neurons, dorsal raphe nucleus (DRN) serotonergic neurons, and locus coeruleus (LC) noradrenergic neurons—yields limited alterations in sleep (*Lu et al., 2006b*; *Chen et al., 2016*; *Fuller et al., 2011*). Additionally, the lateral hypothalamic area (LH), parabrachial nucleus (PB), tuberomammilary nucleus (TMN), paraventricular nucleus of the thalamus (PVT), ventral tegmental area (VTA), and supramammillary nucleus (SUM) have also been demonstrated to be involved in arousal regulation (*Lu et al., 2006b*; *Pedersen et al., 2017*; *Ren et al., 2018*; *Eban-Rothschild et al., 2016*; *Chemelli et al., 1999*; *Sherin et al., 1996*). However, among these wake-promoting nuclei, only LH orexinergic and PB glutamatergic neurons have been shown to be related to hypersomnia. Dysfunction of orexinergic neurons in the LH results in narcolepsy and sleep fragmentation (*Lu et al., 2006b*; *Chemelli et al., 1999*; *Gerashchenko et al., 2003*; *Adamantidis et al., 2007*); PB glutamatergic neurons are considered to serve as a hub, as they receive afferent chemosensory information and play a role in triggering hypercapnia-induced arousal in obstructive sleep apnea (OSA), whereas ablation of PB glutamatergic neurons decreases hypercapnia-induced arousal (*Guyenet and Bayliss, 2015*; *Yokota et al., 2015*; *Kaur et al., 2013*). The further amazing research found that ablation of LH orexinergic neurons and the lateral PB (LPB) have little effect on sleep under baseline conditions, and deletion of the medial PB (MPB) causes only a modest (approximately 20%) reduction in wakefulness (*Fuller et al., 2011*; *Gerashchenko et al., 2003*). Clinically, patients with Parkinson's disease (PD), Alzheimer's disease (AD), Kleine-Levin Syndrome, and idiopathic hypersomnia (IH), in which LH orexinergic and PB glutamatergic neurons are thought to function normally, still show hypersomnolence (*Bollu et al., 2018*). Collectively, these results suggest that the key hypersomnia-controlling nucleus remains unidentified.

Here, we found that the paraventricular nucleus of the hypothalamus (PVH) showed higher c-fos expression during the active period (23:00) than during the inactive period (11:00). The PVH is located in the ventral diencephalon adjacent to the third ventricle (*Qin et al., 2018*). More than 90 % of the PVH consists of glutamatergic neurons expressing vesicle glutamate transporter 2 (vglut2), whereas GABAergic neurons are more scarcely represented (*Vong et al., 2011*; *Xu et al., 2013*; *Ziegler et al., 2005*). $PVH^{vglut2}$ neurons co-express corticotropin-releasing hormone (CRH) (*Zhang et al., 2020*), oxytocin (OT) (*Hung et al., 2017*), and prodynorphin (PDYN) (*Li et al., 2019*).

In the present study, we applied cutting-edge techniques to transgenic mice and found that $PVH^{vglut2}$ neurons were preferentially active during wakefulness, and that the activation of $PVH^{vglut2}$, $PVH^{CRH}$, $PVH^{OT}$, and $PVH^{PDYN}$ neurons induced wakefulness. Conversely, ablation or suppression of $PVH^{vglut2}$ neurons caused hypersomnia-like behaviors. Furthermore, photostimulation of $PVH^{vglut2}$→PB/ventral lateral septum (LSv) circuits immediately drove transitions from non-rapid eye movement (non-REM, NREM) sleep to wakefulness. Taken together, our findings indicate that the PVH is essential for physiologic arousal and the pathogenesis underlying HD.

## Results

### The PVH receives direct inputs from the PVT and PB

To test whether the PVH is one of the key arousal-controlling centers, we visualized an unbiased map of c-fos expression in the PVH after a period of wakefulness or sleep in mice. Immunohistochemistry results showed a 306 % higher number of c-fos-positive cells in the PVH from mice sacrificed in the active period (23:00) compared with those sacrificed in the inactive period (11:00) (*Figure 1A and B*), indicating that the PVH might be an important wake-promoting nucleus. We next determined upstream pathways of the PVH to explore potential connections with identified wake-promoting brain regions. Considering that the PVH contains mainly glutamatergic neurons, we used Cre-dependent rabies virus–mediated monosynaptic retrograde tracing in $vglut2^{Cre}$ mice (*Figure 1C*) to figure out the direct inputs of $PVH^{vglut2}$ neurons. Rabies virus infection was clearly detectable at the infection site in $vglut2^{Cre}$ mice compared with wild-type littermates, indicating no leakage of viral infection (*Figure 1D*). As shown in *Figure 1E* and *Figure 1—figure supplement 1*, we found that $PVH^{vglut2}$ neurons received direct inputs from many regions. Among them, the PVT and PB are considered as two important wake-promoting nuclei; they exhibit higher activities during wakefulness than during

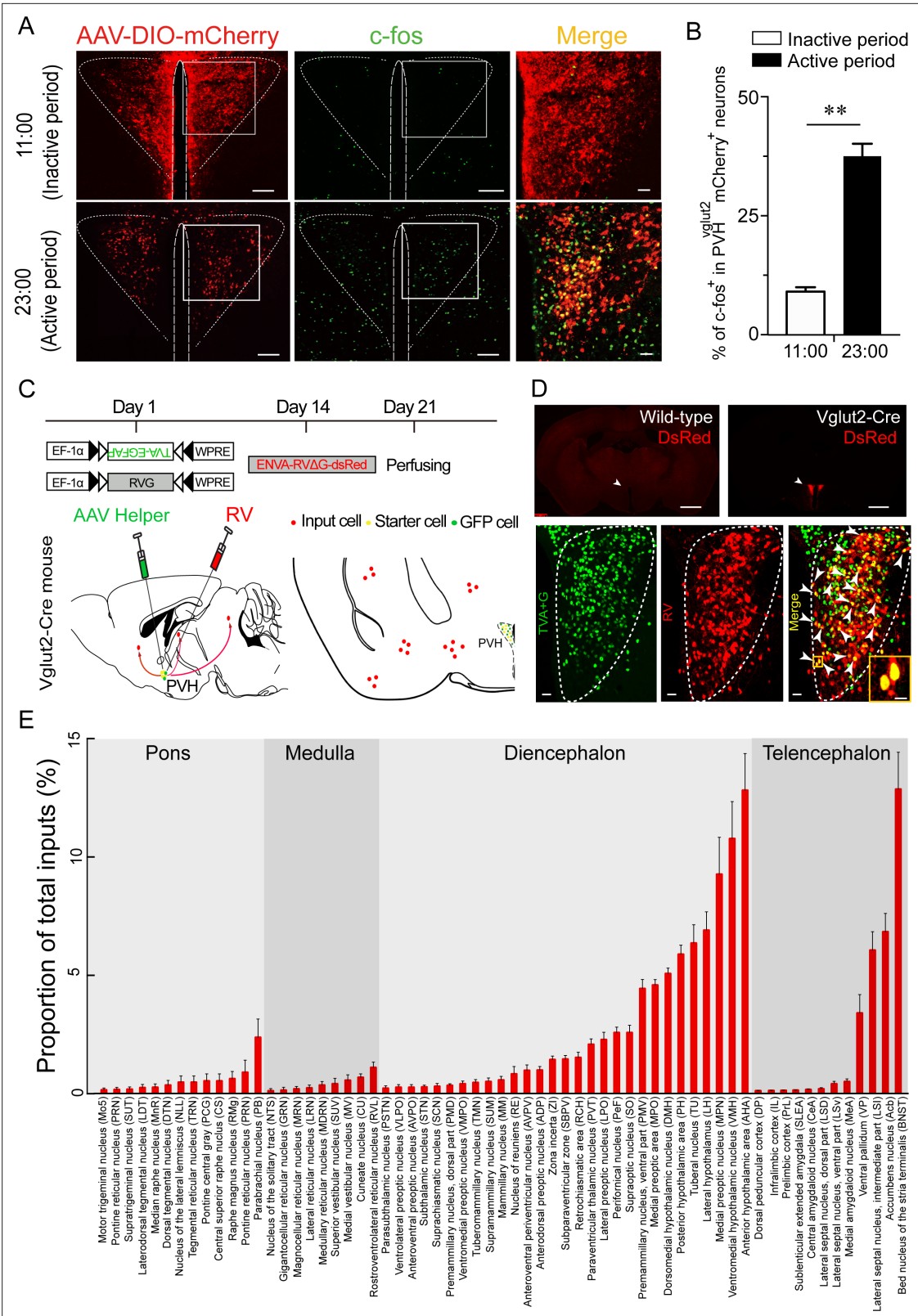

**Figure 1.** C-fos expression during the inactive period versus during the active period and RV retrograde tracing in PVH[vglut2] neurons. (**A**) Representative images showing co-expression of mCherry in the PVH[vglut2] neurons (red) with c-fos immunostaining (green). The merged image is shown in the right panel (yellow). Scale bars: 200 μm. (**B**) The percentage of c-fos[+] neurons in PVH[vglut2] mCherry[+] neurons during the inactive period (11:00) versus during the active period (23:00) (n = 5, unpaired *t* test). Data represented as mean ± SEM (**p < 0.01). (**C**) Experimental procedure of RV retrograde tracing

*Figure 1 continued on next page*

Figure 1 continued

in PVH$^{vglut2}$ neurons. Top: experimental timeline for injecting Cre-dependent helper viruses into the PVH of vglut2$^{Cre}$ mice. Bottom left: a schematic of unilateral microinjection into the PVH in vglut2$^{Cre}$ mice; Bottom right: a schematic drawing showing a brain section of the anatomical site and viral infection in the PVH. (**D**) Top: typical fluorescence images showing RV only labelled in vglut2$^{Cre}$ mice (top left) rather than in wild-type mice (top right). Scale bar: 500 μm. White arrows indicate the PVH area. Bottom: Fluorescence images showing that the starter cells (yellow) labeled by both helper viruses (green) and RV (red) were restrictedly infected in the unilateral PVH in a vglut2$^{Cre}$ mouse. Scale bar: 50 μm. Highly magnified PVH areas with boxes were enlarged and merged in the right panel. Scale bars: 20 μm. (**E**) Percentage of whole-brain, monosynaptic inputs to PVH$^{vglut2}$ neurons. Red columns with different length represent the proportion (or intensity) of inputs. Data represented as mean ± SEM (n = 4 mice).

The online version of this article includes the following figure supplement(s) for figure 1:

**Source data 1.** C-fos expression and RV retrograde tracing of PVH$^{vglut2}$ neurons.

**Figure supplement 1.** Monosynaptic inputs to PVH$^{vglut2}$ neurons.

sleep, and their activation induces rapid transitions from sleep to wakefulness (*Ren et al., 2018*; *Qiu et al., 2016*; *Xu et al., 2021*). Therefore, we speculated that PVH$^{vglut2}$ neurons might act as a crucial node for the control of wakefulness.

## PVH$^{vglut2}$ neurons are preferentially active during wakefulness

To investigate the real-time activities of PVH$^{vglut2}$ neurons across the spontaneous sleep–wake cycles of freely moving mice, we performed in vivo fiber photometry to investigate the real-time activities of PVH$^{vglut2}$ neurons across spontaneous sleep–wake cycles in freely moving mice. The recording mode for fiber photometry and the expression of the Cre-dependent adeno-associated viruses (AAVs) expressing the fluorescent calcium indicator, GCaMP6f (AAV-EF1α-DIO-GCaMP6f), in the PVH of vglut2$^{Cre}$ mice are shown in *Figure 2A and B*. PVH$^{vglut2}$ neuronal activities during wakefulness were significantly higher than those during NREM sleep (*Figure 2C–E*). We next performed in vivo multi-channel electrophysiological recordings to monitor the spike firing of individual PVH neurons in freely behaving mice (*Figure 2F*). PVH neurons exhibited a higher firing rate during wakefulness than during sleep (*Figure 2G and H*). The neuronal firing rate in the PVH gradually decreased before sleep onset and increased during transitions from sleep to wakefulness (*Figure 2I–K*). At the onset of behavioral arousal from NREM sleep, the mean firing rate increased from 4.83 Hz to 13.52 Hz (*Figure 2I*). Collectively, these electrophysiological results clearly indicate a mechanistic framework for the activity-dependent participation of PVH neurons in the regulation of sleep and wakefulness.

## Activation of PVH$^{vglut2}$ neurons significantly increases wakefulness

Next, we investigated the activation effect of PVH$^{vglut2}$ neurons in freely moving mice on wakefulness by injecting AAV-EF1α-double-floxed inverse-orientation (DIO)-hM3Dq-mCherry into the PVH (*Figure 3A and B*). At the beginning of the light phase (zeitgeber time 2 [ZT2]; 9:00), chemogenetic activation of PVH$^{vglut2}$ neurons caused a potent increase in wakefulness lasting approximately 9 hr and concomitantly decreased both NREM and REM sleep (*Figure 3C*). CNO administration (3 mg/kg) resulted in a 140.0 % increase in total wakefulness, as well as 81.6% and 94.5% reduction in NREM and REM sleep, respectively, during the 9 hr post-injection period (*Figure 3D*). However, there was no sleep rebound followed the long-lasting wakefulness, as indicated by no change in the time spent in NREM sleep during the following dark period (19:00–07:00; *Figure 3E*). Compared with vehicle injection, chemogenetic activation of PVH$^{vglut2}$ neurons significantly increased EEG low delta power (0.25–1.00 Hz) and decreased high delta power (1.25–4.75 Hz) (*Figure 3F*). There was no significant difference in the EEG power density of NREM sleep during the day (7:00–18:00) following the day of CNO injection (*Figure 3G*). Similarly, CNO injection during the dark period also significantly increased wakefulness (*Figure 3—figure supplement 1*), further demonstrating that activation of PVH$^{vglut2}$ neurons prolonged arousal even during the dark (active) period.

In order to delineate the potential changes in autonomic function, we measured the heart rate and temperature of PVH$^{vglut2-M3}$ mice after treatment with vehicle or CNO with implantable telemetry devices. We found that chemogenetic activation of PVH$^{vglut2}$ neurons induced an increase in heart rate and temperature for 3 hr, both of which peaked at about 2 hr after injection and returned to the baseline level at about 3 hr following administration of CNO. As for the endocrine function, we detected serum CRH and corticosterone (CORT) levels in PVH$^{vglut2-M3}$ mice after treatment with vehicle or CNO. As shown in *Figure 3—figure supplement 2C*, chemogenetic activation of PVH$^{vglut2}$ neurons

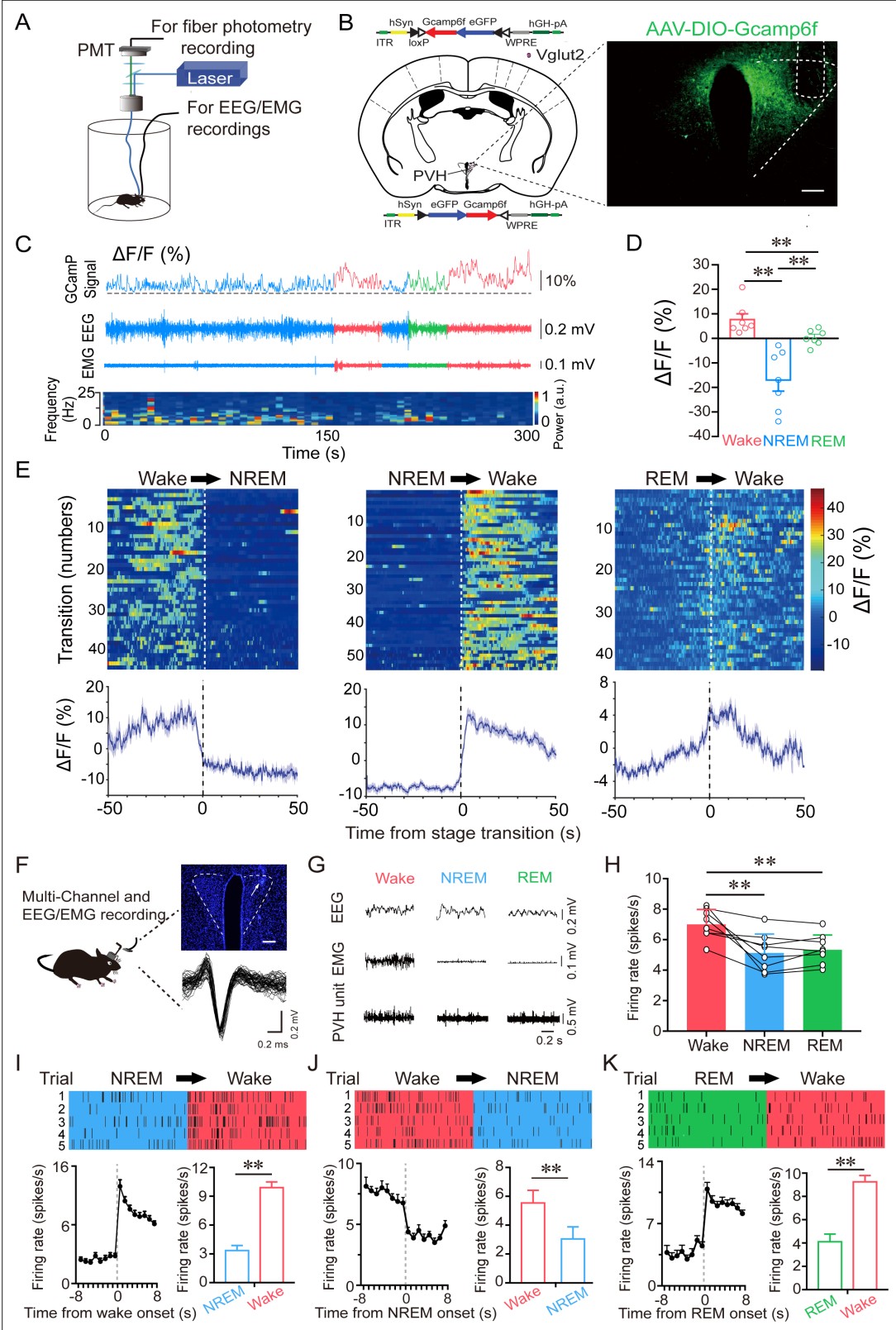

**Figure 2.** PVH^vglut2 neurons are preferentially active during wakefulness. (**A**) Schematic of the fiber photometry setup and in vivo recording configuration (DM dichroic mirror, PMT photomultiplier tube). (**B**) Unilateral viral targeting of AAV-EF1α-DIO-GCaMP6f into the PVH, in which the tip of the fiber optic is above the PVH. Scale bar: 200 μm. (**C**) Representative fluorescent traces, relative EEG power, and EEG/EMG traces across spontaneous sleep–wake states. ΔF/F represents the change in fluorescence from the median of the entire time series. (**D**) Fluorescence (mean ± SEM) during wakefulness, NREM

*Figure 2 continued on next page*

*Figure 2 continued*

sleep, and REM sleep from three mice; the fluorescent signal was the highest during wakefulness, intermediate during REM, and the lowest during NREM sleep (7 sessions from 3 mice, one-way ANOVA followed by Tukey's post-hoc tests; $F_{6,12}$ = 2.94, p< 0.001; p [wake vs NREM] < 0.001, p [wake vs REM] < 0.001, p [NREM vs REM] = 0.013). (**E**) Fluorescent signals aligned to sleep–wake transitions. Upper panel: Individual transitions with color-coded fluorescent intensities (NREM to wake, n = 54; wake to NREM, n = 45; REM to wake, n = 44). Lower panel: Mean (blue trace)± SEM (gray shading) showing the average calcium transients from all the transitions. (**F**) Schematic configuration of in vivo multichannel electrophysiological recordings. Upper panel: A brain slice from a mouse with electrodes implanted in the PVH. White arrow indicates the electrode track. Scale bar: 200 µm. Lower panel: Waveforms from a recorded PVH neuron. (**G**) EEG/EMG and PVH multi-unit recording traces during wakefulness, NREM sleep, and REM sleep. (**H**) Average firing rates of PVH neurons during each state (n = 8 cells from 3 mice, one-way repeated-measures ANOVA followed by LSD post hoc tests; $F_{2,14}$ = 12.51, p [NREM vs wake] < 0.001, p [wake vs NREM] < 0.01, p [NREM vs REM] = 0.613). (**I–K**) Firing rates of PVH neurons during state transitions: NREM to wake (**I**), wake to NREM (**J**), and REM to wake transitions (**K**). Top: Example rastergrams of a PVH neuron during five trials of different state transitions. Bottom left: Average firing rate during the state-transition period. Bottom right: Average firing rate during 8 s before and after state transitions (p [NREM to wake] < 0.01, p [wake to NREM] < 0.01, p [REM to wake] < 0.01, paired t test). Data represented as mean ± SEM ( **p < 0.01).

The online version of this article includes the following figure supplement(s) for figure 2:

**Source data 1.** In vivo fiber photometry and multichannel electrophysiological recordings of PVH^vglut2 neurons.

significantly increased the CRH level at 2, 3, and 4 hr after injection, while the CORT level of the CNO group was only higher than that of the vehicle group at 3 hr following CNO administration.

Considering the potent wake-promoting effect of PVH^vglut2 neurons and the millisecond-scale control of neuronal activity through optogenetic manipulation, we next employed optogenetic methods to elucidate the causal role of the PVH^vglut2 neurons in controlling wakefulness. We stereotaxically injected AAVs expressing channelrhodopsin-2 (AAV-DIO-ChR2-mCherry) into the PVH (*Figure 4A*). Functional expression of ChR2 was verified by in vitro electrophysiology (*Figure 4B*). Next, we applied optical blue-light stimulation (10 ms, 20 Hz, 20–30 mW/mm²) after the onset of stable NREM or REM sleep during the light phase (*Figure 4C*). Optical stimulation of PVH^vglut2 neurons during NREM sleep reliably induced transitions to wakefulness in a frequency-dependent manner (*Figure 4D*). Analysis of the probability of transitions between each pair of sleep–wake states showed that optical stimulation significantly enhanced the probability of wakefulness, along with a complementary decrease in the probability of NREM or REM sleep (*Figure 4E*). To test whether these neurons also contributed to the maintenance of wakefulness, photostimulation was given for 1 hr during the light period (09:00–10:00). Sustained activation of PVH^vglut2 neurons via semi-chronic optical stimulation (10 ms blue-light pulses at 20 Hz for 25 s, every 60 s for 1 hr) significantly increased the amount of wakefulness in ChR2-mCherry mice compared with that of the baseline control between 09:00 and 10:00 (12.3 ± 1.8 min at baseline vs. 48.6 ± 1.5 min after stimulation, n = 5; *Figure 4F*). These findings demonstrate that optogenetic activation of PVH^vglut2 neurons potently enhanced both the initiation and maintenance of wakefulness.

## PVH^vglut2 neurons promote wakefulness via PB and LSv connections

We next sought to determine the downstream targets via which PVH^vglut2 neurons promote wakefulness. Specifically, AAV-hSyn-DIO-eGFP constructs were injected into the PVH of vglut2^Cre mice (*Figure 5—figure supplement 1A,B*). We found that PVH^vglut2 neurons projected to diverse neuroanatomical sites (*Figure 5—figure supplement 1C*, 1*Figure 5—source data 1*). Among them, the PB, PVT and nucleus of the solitary tract (NTS) have been demonstrated to be essential in controlling wakefulness (*Ren et al., 2018*; *Qiu et al., 2016*; *Xu et al., 2021*) and the PB innervates PVH^vglut2 neurons to form bidirectional connections. In addition, the PVH^vglut2-LSv pathway is involved in the regulation of feeding, which is a wakefulness-required behavior (*Xu et al., 2019*). Therefore, we next explored these four pathways to identify the neuronal circuits mediating the wake-promoting effect of PVH^vglut2 neurons. ChR2 was expressed in the PVH with optic fibers targeting terminals in the PB or LSv (*Figure 5A and E*). Optogenetic stimulation (10 ms pulses at 10 Hz for 2 s) of the ChR2-expressing PVH terminals evoked excitatory postsynaptic currents (EPSCs) in most of the patch-recorded PB (n = 6 cells, *Figure 5B*) or LSv (n = 8 cells, *Figure 5F*) neurons. Moreover, 20 Hz stimulation of the bilateral PB or LSv induced a shorter transition from NREM sleep to wakefulness (latency for PB: 1.0 ± 0.8 s, latency for LSv: 1.2 ± 0.9 s) compared with that of the control (*Figure 5C and G*). Analysis of the probability of transition between each pair of sleep–wake states showed that optical stimulation significantly enhanced the probability of wakefulness, along with a complementary decrease in the

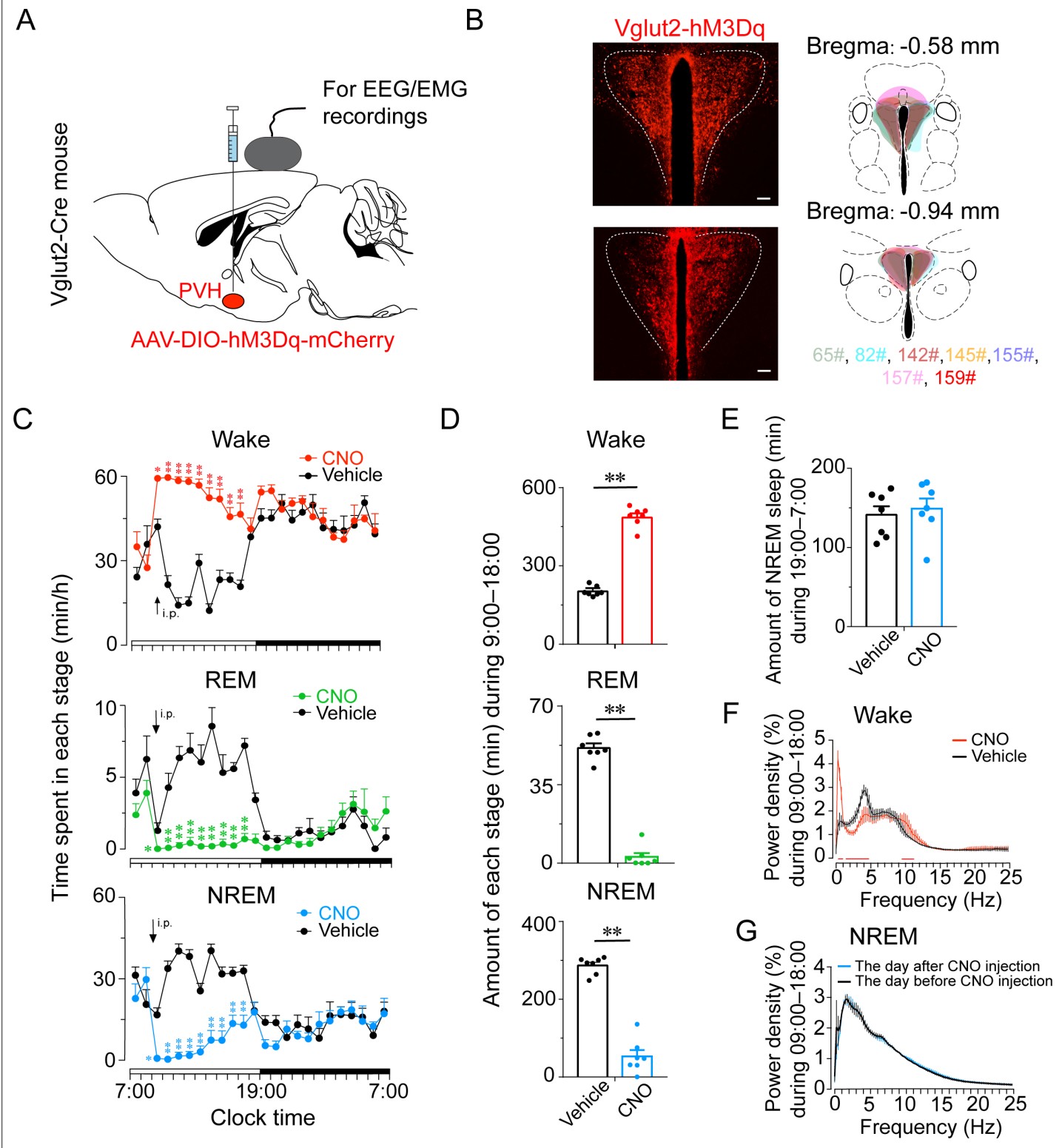

**Figure 3.** Chemogenetic activation of PVH[vglut2] increases wakefulness. (**A, B**) Expression of the AAV injection site in the PVH of vglut2[Cre] mice. Drawings of overlay mCherry expressing sites in the PVH of vglut2[Cre] mice (n = 7, indicated with different colors). (**C**) Time-course changes in wakefulness, NREM sleep, and REM sleep after administration of vehicle or CNO in mice expressing hM3Dq in PVH[vglut2] neurons (n = 7, repeated-measures ANOVA; $F_{1,12}$ = 87.09 [wake], 63.61 [NREM], 612.30 [REM]; p < 0.001 [wake], p < 0.001 [NREM], p < 0.001 [REM]). (**D**) Total time spent in each stage after vehicle or CNO injection into vglut2[Cre] mice (n = 7, paired t test; p < 0.001 [wake], p < 0.001 [NREM], p < 0.001 [REM]). (**E**) Total time spent in NREM sleep during the dark period after vehicle or CNO injection (n = 7, p > 0.05, paired t test). (**F**) EEG power density of wakefulness during 9 hr after vehicle or CNO injection

*Figure 3 continued on next page*

Figure 3 continued

(n = 5; p < 0.05, paired t test). (**G**) EEG power density of NREM sleep during the day (7:00–18:00) before/after the day of CNO injection (n = 5, p > 0.05, paired t test). Data represented as mean ± SEM (*p < 0.05, **p < 0.01, n.s. means no significant difference).

The online version of this article includes the following figure supplement(s) for figure 3:

**Source data 1.** Time spent in each stage of PVH$^{vglut2-M3}$ mice after administration of CNO or saline during the light phase.

**Source data 2.** Time spent in each stage of PVH$^{vglut2}$ mice after administration of CNO or saline during the dark phase.

**Source data 3.** Heart rate, temperature, serum CRH and CORT levels of PVH$^{vglut2-M3}$ mice.

**Figure supplement 1.** Chemogenetic activation of PVH$^{vglut2}$ neurons during the dark phase increases wakefulness.

**Figure supplement 2.** Autonomic and endocrine changes after chemogenetic activation of PVH$^{vglut2}$ neurons.

probabilities of NREM and REM sleep (**Figure 5D and H**). However, optogenetic activation of the other two pathways— PVH$^{vglut2}$→NTS and PVH$^{vglut2}$→PVT pathways did not alter sleep—wake states (**Figure 5—figure supplement 2**). These results demonstrated that PVH$^{vglut2}$→PB and PVH$^{vglut2}$→LSv circuits mediated the wakefulness-controlling effect of PVH$^{vglut2}$ neurons.

## PVH$^{vglut2}$ neurons exert wakefulness via PVH$^{OT}$, PVH$^{PDYN}$, and PVH$^{CRH}$ neurons

We further explored the arousal-promoting roles of subtype neurons of PVH$^{vglut2}$ neurons (PVH$^{OT}$, PVH$^{PDYN}$ and PVH$^{CRH}$ neurons) by injecting AAV-DIO-hM3Dq-mCherry into the PVH of OT$^{Cre}$ mice, PDYN$^{Cre}$ mice and CRH$^{Cre}$ mice, respectively (**Figure 6A, D and G**). The locations of hM3Dq expression in PVH$^{OT}$, PVH$^{PDYN}$ and PVH$^{CRH}$ neurons were verified after mice sacrifice (**Figure 6B, E and H**). We found that chemogenetic activation either PVH$^{OT}$ or PVH$^{PDYN}$ neurons both induced increased wakefulness (49.4% and 75.8%, respectively) and decreased NREM sleep (53.9% and 88.5%, respectively) that lasted for 1 hr (**Figure 6C and F**). Similarly, chemogenetic activation of PVH$^{CRH}$ neurons caused a potent increase in wakefulness lasting approximately 3 hr and concomitantly decreased both NREM and REM sleep. CNO administration (3 mg/kg) induced a 75.8 % increase in wakefulness and a 67.7% and 46.3% reduction in NREM and REM sleep, respectively, during the 3 hr post-injection period (**Figure 6I**).

## Inhibition or ablation of PVH$^{vglut2}$ neurons induces hypersomnia-like behaviors

To determine whether PVH$^{vglut2}$ neurons are necessary for natural wakefulness, we inhibited PVH$^{vglut2}$ neurons with AAV constructs encoding engineered Gi-coupled hM4D receptor (AAV-EF1α-DIO-hM4Di)-mCherry (**Figure 7A and B**). Chemogenetic inhibition of PVH$^{vglut2}$ neurons induced a 3 hr increase in NREM sleep compared with vehicle injection. At the beginning of the dark phase (ZT14; 21:00), CNO injection resulted in a 64.0 % increase in NREM sleep during the 5 hr post-injection period, which was accompanied by a 26.0 % decrease in wakefulness (**Figure 7C**). These results suggested that the PVH$^{vglut2}$ neurons play a critical role in sleep—wake regulation.

To further assess the functional importance of PVH$^{vglut2}$ neurons controlling physiological wakefulness, we specifically ablated these neurons by bilaterally microinjecting AAV-EF1a-DIO-taCasp3-TEVp into the PVH region of vglut2$^{Cre}$ mice. This construct expressed a designer pro-caspase-3 (pro-taCasp3) in the PVH, the activation of which caused apoptosis (**Figure 7D and E**). Compared with the control group, mice that underwent PVH$^{vglut2}$ neuronal ablation showed a 28.6 % decrease in the amount of wakefulness and a 74.7 % increase in the amount of NREM sleep during the dark period. Similarly, ablation of PVH$^{vglut2}$ neurons induced a 20.8 % reduction in wakefulness and 30.6 % increase in NREM sleep across an entire 24 hr light/dark cycle (**Figure 7F**). These results indicate that PVH$^{vglut2}$ neurons are necessary for wakefulness regulation under physiological conditions, and that dysfunction of these neurons may induce hypersomnia.

## Discussion

Adequate wakefulness is essential for life and survival. In the present study, we identified the PVH as a critical hypothalamic nucleus for the pathogenesis underlying HD. In previous studies, a 15 %

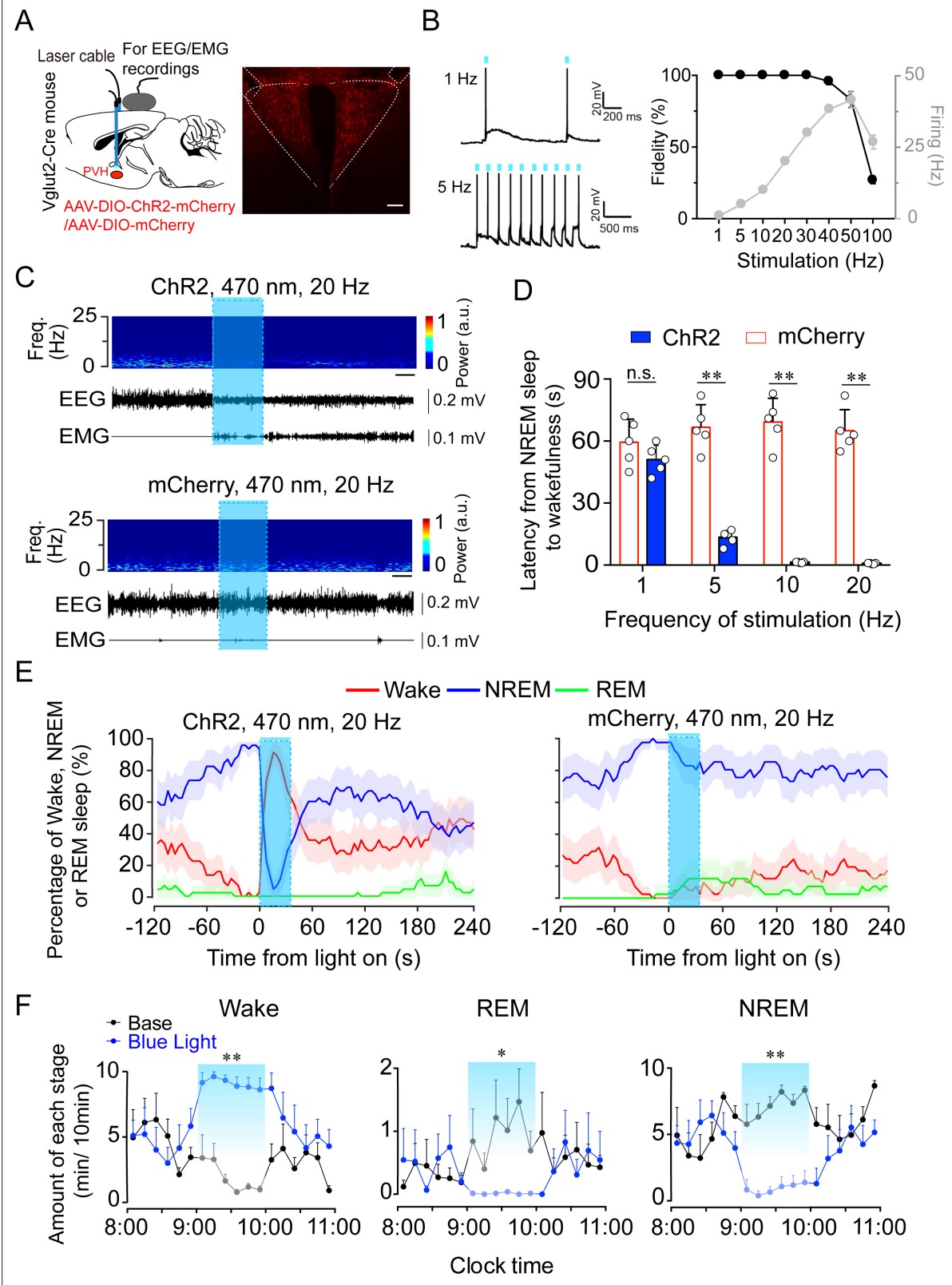

**Figure 4.** Optogenetic activation of PVH[vglut2] neurons induces a rapid transition from NREM sleep to wakefulness. (**A**) Left: Schematic of optogenetic manipulation of PVH[vglut2] neurons and EEG/EMG recordings. Right: ChR2-mCherry expression and location of optical fiber in the PVH. Scale bar: 200 μm. (**B**) Example traces (left) and fidelity of action potential firing (right) of ChR2-expressing PVH neurons evoked by 473 nm light stimulation with different frequencies. (**C**) Representative EEG/EMG traces, and heatmap of EEG power spectra showing that acute photostimulation (20 Hz/10 ms) applied

*Figure 4 continued on next page*

*Figure 4 continued*

during NREM sleep induced a transition to wakefulness in a ChR2-mCherry mouse. Scale bar: 10 s. (**D**) Latencies of transitions from NREM sleep to wakefulness after photostimulation at different frequencies (n = 5, unpaired t test; 1 Hz, $t_8$ = 1.4, p = 0.19; 5 Hz, $t_8$ = 10.29, p < 0.01; 10 Hz, $t_8$ = 13.3, p < 0.01; 20 Hz, $t_8$ = 14.04, p < 0.01). (**E**) Sleep stage after blue-light stimulation in a PVH-vglut2-ChR2 mouse or PVH-vglut2-mCherry mouse. Percentages of NREM, REM, and wakefulness during short-stimulation experiments. (**F**) Time course during semi-chronic optogenetic experiments (20 Hz/10 ms, 25 s on /35 s off). The blue column indicates the photostimulation period of the stimulation group (n = 5, repeated-measures ANOVA; $F_{1,8}$ = 59.37 (wake), 18.20 (REM), 103.30 (NREM); p < 0.001 [wake], p = 0.003 [REM], p < 0.001 [NREM]). Data represented as the mean ± SEM (*p < 0.05, **p < 0.01).

The online version of this article includes the following figure supplement(s) for figure 4:

**Source data 1.** Latencies of transitions and time course in semi-chronic optogenetic experiments of PVH$^{vglut2-ChR2}$ mice.

reduction in baseline wakefulness is considered significant (*Lu et al., 2006b*; *Lu et al., 2006a*). Lu et al have reported that lesions of the PPT and the ventral sublaterodorsal nucleus (vSLD) result in a 20–30% reduction in baseline wakefulness (*Lu et al., 2006b*). However, bidirectional chemogenetic manipulations that inhibit the PPT or activate SLD neurons have been shown to have little influence on baseline sleep (*Kroeger et al., 2017*; *Erickson et al., 2019*). In the present study, we found that ablation of PVH$^{vglut2}$ neurons in mice induced a 30.6 % reduction in wakefulness across the 24 hr light/dark cycle, highlighting the significance of PVH$^{vglut2}$ neurons in maintaining wakefulness and preventing hypersomnia. Furthermore, in our murine experiments, no sleep rebound was seen after PVH$^{vglut2}$-activation-induced enhancement of wakefulness. This finding is in accordance with previous studies using chemogenetics to specifically activate wake-promoting neuronal populations (*Pedersen et al., 2017*; *Erickson et al., 2019*; *Anaclet et al., 2015*; *Venner et al., 2016*) and indicates that chemogenetic activation of wake-promoting neuronal populations does not enhance the homeostatic drive for sleep. Taken together, our present findings provide evidence of the sufficient and necessary wake-promoting action of PVH$^{vglut2}$ neurons in preventing hypersomnia.

The PVH is composed of abundant, diverse, and functionally distinct groups of neuroendocrine neurons, including CRH, OT, and PDYN neurons (*Zhang et al., 2020*; *Hung et al., 2017*; *Sterley et al., 2018*; *Li et al., 2019*). The PVH is estimated to consist of approximately 56,000 neurons in humans (*Morton, 1969*), of which 25,000 neurons express OT (*Wierda et al., 1991*), 21,000 neurons express PDYN (*Li et al., 2019*; *Purba et al., 1993*; *Purba et al., 1995*) and 2000 neurons express CRH (*Xu et al., 2020*). Almost 100 % of neurons expressing CRH, PDYN and OXT were found to co-express with vglut2 in the middle PVH and posterior PVH, which were also the main distribution regions of these three neuropeptides populations in the PVH (*Ziegler et al., 2005*; *Holmes et al., 1986*). PVH$^{CRH}$ neurons regulate stress, fear, and the immune response, as well as neuroendocrine and autonomic functions (*Zhang et al., 2020*; *Xu et al., 2019*; *Kondoh et al., 2016*; *Mezey et al., 1984*; *Winter and Jurek, 2019*). There is mounting evidence that exposure to various stressors induces CRH and OT release into the peripheral circulation (*Yuan et al., 2019*). However, OT exhibits some opposing actions to those of CRH. CRH serves as the starting point and main driver of the hypothalamic-pituitary-adrenal (HPA) axis (*Zhang et al., 2020*; *Füzesi et al., 2016*; *Windle et al., 2004*), while OT inhibits the activity of the HPA axis (*Daviu et al., 2020*). PVH$^{CRH}$ neurons orchestrate stress-related behaviors, such as grooming, fear, rearing, and walking (*Windle et al., 2004*; *Xiao et al., 2017*), while PVH$^{OT}$ neurons modulate reward circuits and play a role in mitigating the stress response (*Yuan et al., 2019*; *Ono et al., 2020*). In the present study, our results showed that PVH$^{CRH}$, PVH$^{OT}$, and PVH$^{PDYN}$ neurons play important roles in the regulation of wakefulness. Considering that PVH$^{CRH}$ neurons play a role in the circadian regulation of wakefulness (*Li et al., 2020a*) and optogenetic stimulation of PVH$^{CRH}$ neurons can simulate stress-induced insomnia (*Spencer and Deak, 2017*), our results provide further evidence that PVH$^{CRH}$ neurons play a role in stress-related insomnia. PVH$^{PDYN}$ neurons are key regulators of satiety, and they project to the PB (*Li et al., 2019*). We found that chemogenetic activation of PVH$^{PDYN}$ neurons induced short-duration wakefulness; hence, dysfunction of PVH$^{PDYN}$ neurons may represent one of the causes of sleep-related eating disorders .

It is surprising that a single injection of CNO into mice expressing Dq designer receptors exclusively activated by designer drugs (DREADDs) in the PVH can affect sleep for up to 9 hr, but there are several possible mechanisms can explain this. First, given that PVH glutaminergic neurons are largely co-expressed with CRH, PDYN, and OT neurons, which are all wake-promoting neurons, the 9 hr of wakefulness induced by activation of PVH$^{vglut2}$ neurons partly results from activation of these types of neurons. Furthermore, the PVH sends direct projections to the PB, and it was reported that

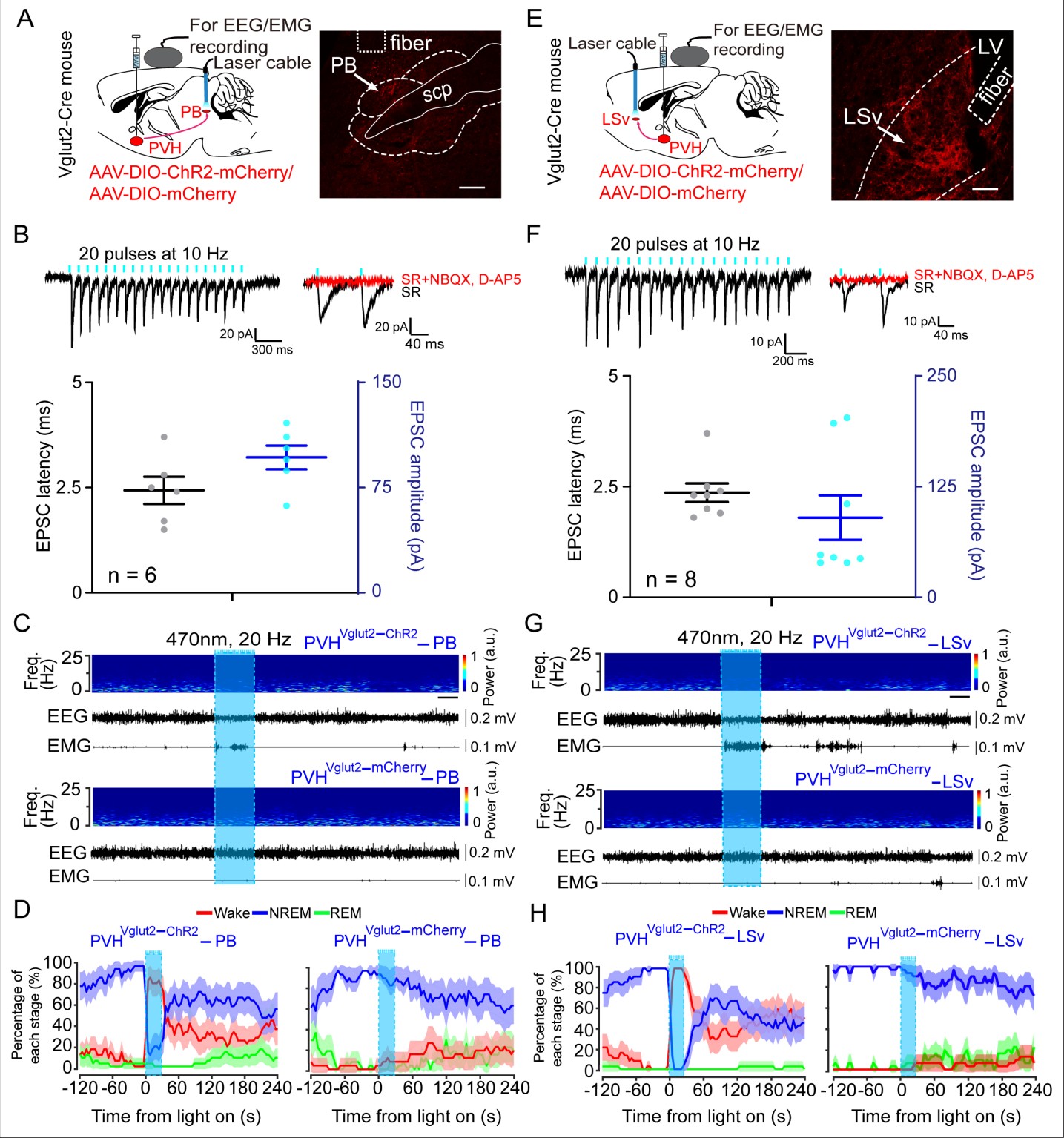

**Figure 5.** PVH^vglut2 neurons control arousal through PB and LSv pathways. (**A, E**) Left: Schematic diagram showing the location of the optic fiber in the PB and LSv, and EEG/EMG recordings of a vglut2^Cre mouse bilaterally injected with AAV-ChR2-mCherry or AAV-mCherry in the PVH. Right: Brain section stained against mCherry to confirm that ChR2 protein expressed in the PVH. Superior cerebellar peduncle, scp. Scale bar: 200 μm. (**B, F**) Upper-left panel: Photostimulation-evoked EPSCs in PB neurons (**B**) and LSv neurons (**F**). Upper-right panel: Photostimulation-evoked EPSCs were completely blocked in the presence of NBQX (20 μM) and D-AP5 (25 μM). Lower panel: Latency (left axis) and amplitude (right axis) of light-evoked EPSCs in PB neurons (**B**) and LSv neurons (**F**). (**C, G**) Representative EEG/EMG traces, and a heatmap of EEG power spectra showing that acute photostimulation

*Figure 5 continued on next page*

*Figure 5 continued*

(20 Hz/10 ms) of PVH-PB (**C**) and PVH-LSv (**G**) pathways during NREM sleep induced a transition to wakefulness in a ChR2-mCherry mouse. Scale bar: 10 s. (**D, H**) Sleep stages after blue-light stimulation of PVH-PB (**D**) and PVH-LSv (**H**) pathways in ChR2-mCherry mice or mCherry control mice.

The online version of this article includes the following source data and figure supplement(s) for figure 5:

**Source data 1.** Statistical analysis of whole-brain outputs from PVH[vglut2] neurons.

**Source data 2.** In vitro photostimulation-evoked EPSCs in PB and LSv neurons.

**Figure supplement 1.** Representative regions with axonal projection from PVH[vglut2] neurons.

**Figure supplement 2.** Optogenetic activation of PVH[vglut2]→NTS or PVH[vglut2]→PVT pathway has no effect on sleep–wake states.

chemogenetic activation of the PB produced ~11 hr of continuous wakefulness in rats (*Qiu et al., 2016*). Moreover, endocrine and autonomic function changes may also contribute to the 9 hr of wakefulness. Glucocorticoid hormones (CORT for mice) are the effector hormones of the HPA axis neuroendocrine system, which are regulated by CRH (secreted by neurons in the medial parvocellular portion of the PVH) and produce direct negative feedback inhibition of CRH neurons in the PVH (*Purba et al., 1994*). Our results showed that although chemogenetic activation of PVH[vglut2] neurons induced higher CRH levels, it did not cause a continuous high level of CORT. These results suggested that changes in glucocorticoid levels indeed occurred but did not last for 9 hr, so the long-lasting effects of wakefulness induced by PVH glutaminergic neurons' activation could only partly be explained by endocrine and autonomic function changes; the wake-promoting neural circuits of PVH[vglut2] neurons played a more important role in the 9 hr of wakefulness.

As our results show, PVH[vglut2] neurons send widespread projections to distinct brain structures. According to previous research, PVH[CRH], PVH[OT], and PVH[PDYN] neurons all project to the median eminence area (ME) (*Xu et al., 2019*; *Li et al., 2019*), which serves as an interface between the neural and peripheral endocrine systems. Consistent with these studies, we also observed GFP-positive fibers in the ME. Our present results show that PVH[vglut2] neurons promote wakefulness via PB and LSv connections. Previous studies reported that the PVH→LSv projection exhibited scalable control over feeding and negative emotional states (*Xu et al., 2019*), and PVH[PDYN]→PB projections are necessary for satiety (*Li et al., 2019*), which are all wakefulness-required behaviors.

HD is one of the most common symptoms in many neurological disorders and mental diseases, including PD, AD, IH, OSA, Huntington's disease, major depressive disorder (MDD), bipolar disorder (BD), Kleine-Levin syndrome, and depression (*Mahowald and Schenck, 2005*; *Bollu et al., 2018*). A robust reduction in the number of PVH neurons has been found in MDD, BD, AD, and PD patients (*Manaye et al., 2005*; *Baloyannis et al., 2015*; *Zhang et al., 2017*). Thus, our present findings provide evidence that dysfunction of the PVH may contribute to the occurrence of hypersomnia in these diseases.

In conclusion, our results indicate that dysfunction of the PVH is crucial for the pathogenesis underlying HD, and PVH[vglut2] and their subtype neurons (PVH[OT], PVH[PDYN], and PVH[CRH]) are critical for wakefulness.

## Materials and methods
### Animals
Vglut2[Cre] (*Slc1716< Cre >* ) mice were obtained from Jackson Laboratory (Bar Harbor, Maine, USA). CRH[Cre] (*Crh< Cre >* ) mice were obtained from the Shanghai Model Organisms Center. PDYN[Cre] (*Pdyn< Cre >* ) mice were generously provided by Wei Shen. OT[Cre] (*Oxt< Cre >* ) mice were kindly provided by Xiao Lei. Mice were housed in a soundproof room at an ambient temperature of 24°C ± 0.5°C, with a relative humidity of 60% ± 2%. A 12 hr light/dark cycle (100 Lux, light on at 07:00) was automatically controlled (*Li et al., 2020b*). Food and water were available ad libitum. Male heterozygous mice at 6–8 weeks of age were used for all experiments. All animal experiments were approved by the Medical Experimental Animal Administrative Committee of Shanghai. All experimental procedures involving animals were approved by the Animal Experiment and Use Committee of Fudan University (20150119–067).

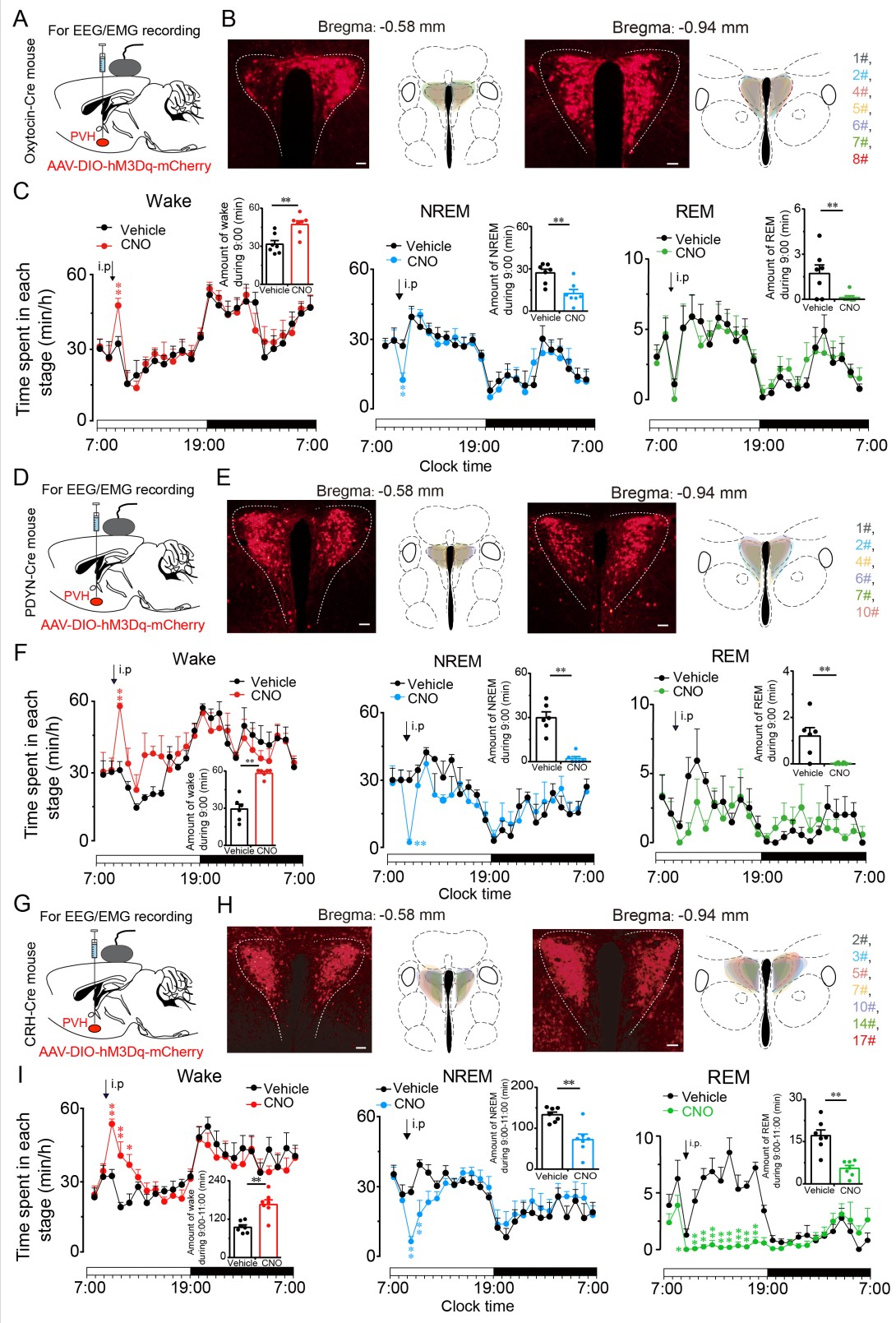

**Figure 6.** PVH^vglut2 neurons exert wakefulness via PVH^OT, PVH^PDYN and PVH^CRH neurons. (**A, D,** G) Schematic drawings of the chemogenetic experiment in OT^Cre mice, PDYN^Cre mice and CRH^Cre mice. (**B, E, H**) Location of hM3Dq expression in PVH^OT, PVH^PDYN, and PVH^CRH neurons. Right panel: Drawings of mCherry expressing sites in the PVH. Scale bars: 200 μm. (**C, F, I**) Time-course changes in wakefulness, NREM sleep, and REM sleep after administration of vehicle or CNO in mice expressing hM3Dq in PVH^OT (**C**, n = 7, repeated-measures ANOVA; $F_{1, 12}$ = 0.41 [wake], 0.74 [NREM], 0.02 [REM]; p < 0.05

*Figure 6 continued on next page*

*Figure 6 continued*

[wake], p < 0.05 [NREM], p < 0.05 [REM]), PVH$^{PDYN}$ neurons (**F**, n = 6, repeated-measures ANOVA; F$_{1, 10}$ = 0.28 [wake], 0.38 [NREM], 0.02 [REM]; p < 0.05 [wake], p < 0.05 [NREM], p < 0.05 [REM]), and PVH$^{CRH}$ neurons (**I**, n = 7, repeated-measures ANOVA; F$_{1, 12}$ = 0.06 [wake], 0.01 [NREM], 0.83 [REM]; p < 0.05 [wake], p < 0.05 [NREM], p < 0.05 [REM]). Inset: Total time spent in each stage after vehicle or CNO injection into mice expressing hM3Dq in PVH$^{OT}$ neurons (**C**, n = 7, paired t test; p = 0.10 [wake], 0.10 [NREM], 0.39 [REM]), PVH$^{PDYN}$ neurons (**F**, n = 6, paired t test, p < 0.01 [wake], p < 0.01 [NREM], p < 0.01 [REM]), and PVH$^{CRH}$ neurons (**I**, n = 7, paired t test; p = 0.003 (wake), p = 0.002 (NREM), p = 0.1 (REM). Data represented as the mean ± SEM (**p < 0.01).

The online version of this article includes the following figure supplement(s) for figure 6:

**Source data 1.** Time spent in each stage of PVH$^{OT-M3}$ mice, PVH$^{PDYN-M3}$ mice, and PVH$^{CRH-M3}$ mice after administration of CNO or saline.

### Preparation of viral vectors

The AAVs of serotype rh10 for AAV-hSyn-DIO-hM3Dq-mCherry, AAV-hSyn-DIO-hM4Di-mCherry, AAV-hSyn-DIO-ChR2-mCherry, AAV-hSyn-DIO-mCherry, AAV-EF1α-DIO-GCaMP6f, AAV-hSyn-DIO-eGFP and AAV-CAG-FLEX-taCasp3-TEVp were used. AAV vectors were packaged into serotype 2/9 vectors, which consisted of AAV2 ITR genomes coupled with AAV9 serotype capsid proteins. The final viral concentrations of the transgenes were in the range of 1–5 × 10$^{12}$ viral particles/mL. Unilateral injections were performed in in vivo fiber photometry (AAV-EF1α-DIO-GCaMP6f), anterograde tracing (AAV-hSyn-DIO-eGFP) experiments, while bilateral injections were used in chemogenetic and optogenetic modulation (AAV-hSyn-DIO-hM3Dq-mCherry, AAV-hSyn-DIO-hM4Di-mCherry, AAV-hSyn-DIO-ChR2-mCherry, AAV-hSyn-DIO-mCherry) and ablation experiments (AAV-CAG-FLEX-taCasp3-TEVp).

### Surgery and injection of viral vectors

According to previous studies (*Luo et al., 2018*; *Zhong et al., 2021*; *Bao et al., 2021*; *Li et al., 2017*), after finishing all related experiments, all mice were anesthetized with chloral hydrate (360 mg/kg, i.p.) for surgical procedures and were placed in a stereotaxic apparatus (RWD, Shenzhen, China). The skin above the skull was cut, a burr hole was made, and a small craniotomy was performed above the PVH. AAV constructs were slowly injected (30 nL/min) into the bilateral PVH (70 nL for each position; AP = −0.5 mm; ML = ± 0.2 mm; DV = −4.2 mm) for polysomnographic recordings and brain-slice electrophysiology, or were unilaterally injected into the PVH for neuronal tracing. The glass pipette was left in the brain for an additional 10 min following injections and was then slowly withdrawn. All mice were implanted with electrodes for EEG and EMG recordings that were used for in vivo tests at four weeks after injections under anesthesia of chloral hydrate ( 360 mg/kg, i.p.). The implant consisted of two stainless steel screws (1 mm in diameter), and EEG electrodes were inserted through the skull ( + 1.5 mm anteroposterior; −2.0 mm mediolateral from bregma or lambda), while two flexible silver wires were inserted into the neck muscles. The electrodes were attached to a mini-connector and were fixed to the skull with dental cement. The scalp wound was sutured, and the mouse was then kept in a warm environment until it resumed normal activity.

### Polysomnographic recordings and analysis

After a 2–3 week recovery period, each mouse was individually housed in a recording chamber and habituated to the recording cable for 2–3 days before electrophysiological recordings. Simultaneous EEG/EMG recordings were carried out with a slip ring so that movement of the mice would not be restricted. For experiments using DREADDs, the recordings started at 07:00 (i.e., at the beginning of the light period), and each mouse received either vehicle or CNO (3 mg/kg, C2041, LKT) treatment for two consecutive days at 09:00 (inactive period) or 21:00 (active period). As previously described (*Ren et al., 2018*; *Zhong et al., 2021*), EEG/EMG signals were amplified and filtered (0.5–30 Hz for EEG, 40–200 Hz for EMG), and were then digitized at 128 Hz and recorded with SleepSign software (Kissei Comtec, Nagano, Japan). Sleep–wake states were automatically classified into 4 s epochs as follows: Wakefulness was considered to have desynchronized EEG and high levels of EMG activity, NREM sleep was considered to have synchronized, high-amplitude, low-frequency (0.5–4 Hz) EEG signals in the absence of motor activity; and REM sleep was considered to have pronounced theta-like (4–9 Hz) EEG activity and muscle atonia. All scoring was automated based on EEG and EMG waveforms in 4 s epochs for both chemogenetic and optogenetic studies.

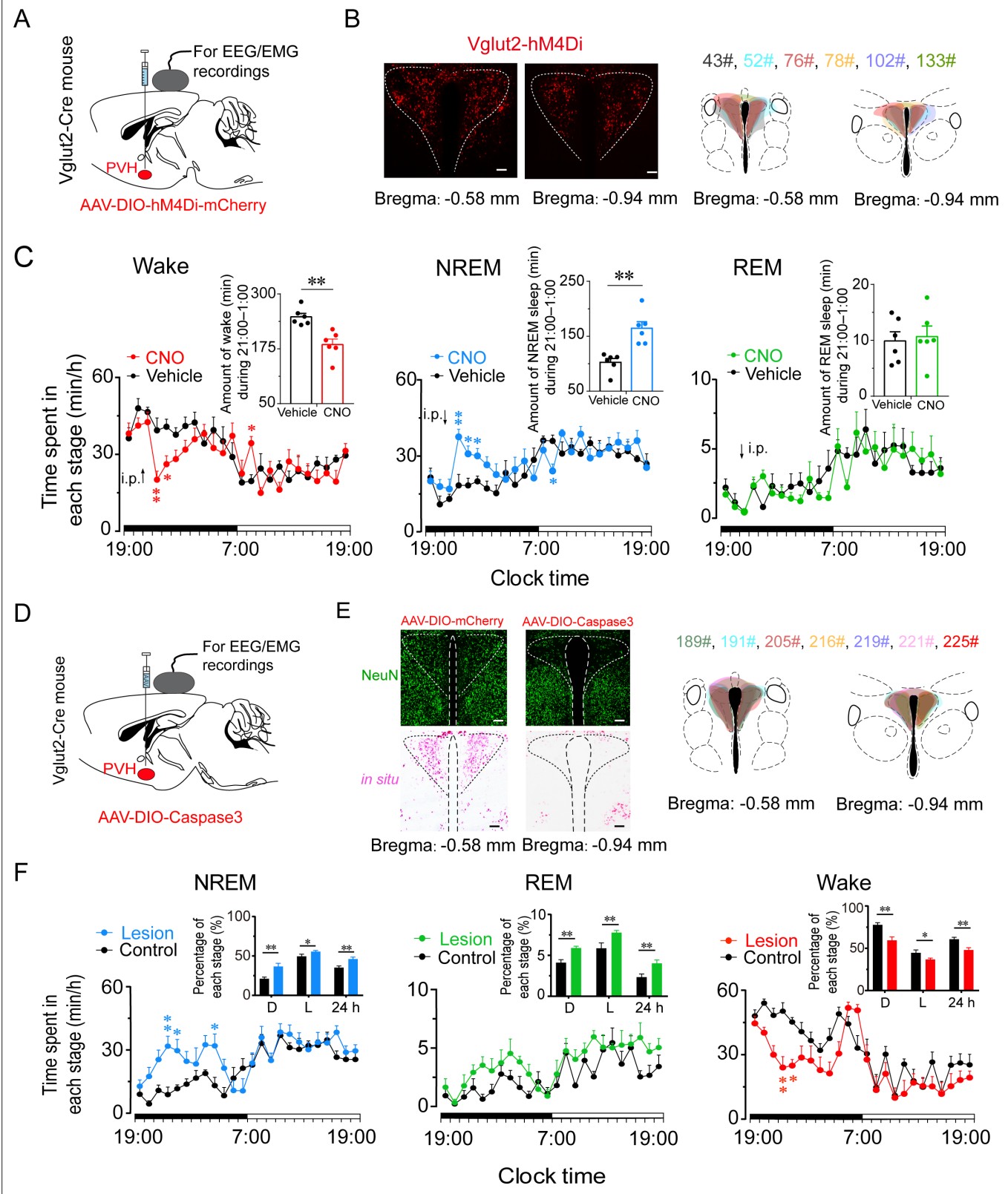

**Figure 7.** Inhibition or ablation of PVH^vglut2 neurons induces hypersomnia-like behaviors. (**A**) Expression of AAV injection site in the PVH of vglut2^Cre mice. (**B**) Left panel: Location of hM4Di expression in the PVH^vglut2 neurons. Right panel: Drawings of overlay mCherry expressing sites in the PVH of vglut2^Cre mice (n = 6, indicated with different colors). Scale bars: 200 μm. (**C**) Time-course changes in NREM sleep, wakefulness, and REM sleep after administration of vehicle or CNO in mice expressing hM4Di in PVH^vglut2 neurons (n = 6, repeated-measures ANOVA; F1,10 = 21.95 [wake], 7.68 [NREM],

*Figure 7 continued on next page*

*Figure 7 continued*

29.23 [REM]). Inset: Total time spent in each stage after vehicle or CNO injection (n = 6, paired t test). (**D**) Expression of an AAV injection site in the PVH of vglut2^Cre mice. (**E**) Left panel: Image showing NeuN (neuron-specific nuclear protein) staining (top) and vglut2 in situ hybridization (bottom) from a control mouse (left) and a mouse with a PVH lesion (right), Scale bars: 200 µm. Right panel: Drawings of superimposed ablation areas in the PVH of vglut2^Cre mice (n = 7, indicated with different colors). (**F**) Time spent in each stage across the 24 hr sleep–wake cycle. L, light phase; D, dark phase. Control group, n = 6; lesion group, n = 7, repeated-measures ANOVA; $F_{1,10}$ = 2.88 (wake), 2.90 (NREM), 0.06 (REM); Inside panel: Percentages in different sleep–wake stages across the 24 hr sleep–wake cycle (unpaired t test; NREM, dark phase, $t_{11}$=3.27, p < 0.01; light phase, $t_{11}$ = 2.32, p = 0.04; 24 h, $t_{11}$ = 3.27, p < 0.01; REM, dark phase, $t_{11}$ = 2.94, p = 0.01; light phase, $t_{11}$=2.88, p = 0.05; 24 hr, $t_{11}$ = 4.36, p < 0.01; wake, dark phase, $t_{11}$ = 3.61, p < 0.01; light phase, $t_{11}$=2.24, p = 0.05; 24 hr, $t_{11}$=3.56, p < 0.01). Data represented as mean ± SEM (*p < 0.05, **p < 0.01).

The online version of this article includes the following figure supplement(s) for figure 7:

**Source data 1.** Time spent in each stage of PVH^vglut2-m4 mice after administration of CNO or saline and time spent in each stage of PVH ^vglut2-Caspase3 mice.

## Optogenetic stimulation

Before the testing day, mice were given one day to adapt to optical fiber cables (0.8 m long, 200 µm diameter; RWD) that were placed inside the implanted fiber cannulae. On the testing day, 473 nm laser pulses (10 ms, 20 Hz) were delivered via an optic cable (Newton Inc, Hangzhou, China) using a pulse generator. Light pulse trains were generated via a stimulator (SEN-7103, Nihon Kohden, Japan) and delivered through an isolator (SS-102J, Nihon Kohden). For acute photostimulation, each stimulation epoch was applied at 20 s after identifying a stable NREM or REM sleep event via real-time online EEG/EMG analysis. Light pulse trains (5 ms pulses of various frequencies and durations) were programmed and conducted during the inactive period. For chronic photostimulation, programmed light pulse trains ( 10 ms blue-light pulses at 20 Hz for 25 s, every 60 s for 1 hr) were used. The 473 nm laser stimulation was performed from 09:00 to 10:00. Baseline EEG/EMG recordings were acquired at the same time of day on the previous day prior to laser stimulation. Sleep–wake cycle parameters (e.g. durations of NREM sleep, REM sleep, and wakefulness, as well as sleep–wake transitions) were scored over an entire hour for each mouse. After receiving photostimulation, mice were sacrificed at 30 min after the final stimulation for subsequent c-fos staining.

## In vitro electrophysiological recordings

At 3–4 weeks after AAV-ChR2 injections, vglut2^Cre mice were anesthetized and transcardially perfused with an ice-cold slicing buffer containing the following (in mM): 213 sucrose, 26 NaHCO₃, 10 glucose, 0.1 CaCl₂, 3 MgSO₄, 2.5 KCl, 1.25 NaH₂PO₄, 2 sodium pyruvate, and 0.4 ascorbic acid. The buffer was saturated with 95 % O₂ and 5 % CO₂. Brains were then rapidly removed, and acute coronal slices (300 µm) containing the PVH were cut using a vibratome (Leica VT 1200 S, Nussloch, Germany). Next, slices were transferred to a holding chamber containing normal recording artificial cerebrospinal fluid (aCSF) containing the following (in mM): 119 NaCl, 26 NaHCO₃, 25 glucose, 2.5 KCl, 2CaCl₂, 1.25 NaH₂PO₄, and 1.0 MgSO₄. After being transferred, slices were allowed to recover for 30 min at 32 °C. Then, slices were maintained at room temperature for at least 30 min before recordings. During recordings, slices were transferred to and submerged in a recording chamber in which oxygenated aCSF was continuously perfused.

Expression of ChR2 was confirmed by visualization of mCherry fluorescence in PVH neuronal somata and axonal terminals. Neurons were identified and visualized with an upright microscope (Olympus, Japan) equipped with differential contrast optics, including a 40× water-immersion objective lens (BX51WI, Olympus). Images were detected with an infrared-sensitive CCD camera (OptiMOS, Q-imaging). Patch-clamp recordings were performed with capillary glass pipettes filled with an intrapipette solution containing the following (in mM): 130 potassium gluconate, 10 KCl, 10 Hepes, 0.5 EGTA, 4 ATP-Mg, 0.5 GTP-Na, and 10 phosphocreatine, adjusting to a pH of 7.2–7.4 with KOH.

Whole-cell patch-clamp recordings were obtained using a MultiClamp 700B amplifier (Molecular Device, Union City, CA, USA) and a Digidata 1,440 A A/D converter (Molecular Device). Signals were sampled at 10 kHz and filtered at 2 kHz. Data were acquired and analyzed using pClamp 10.3 software (pClamp, Molecular Devices). ChR2 stimulation was evoked using 470 nm light. In voltage-clamp experiments, the holding potential was –70 mV. When needed, 20 µM of 2,3-dihydroxy 6-nitro-7-sulfamoyl-benzoquinoxaline-2,3-dione (NBQX, 1044, Tocris Bioscience, UK), 25 µM of d-(-)–2-amino-5-phosphonopentanoic acid (D-AP5, 0106, Tocris Bioscience, UK), and 10 µM of SR95531 (SR, ab144487, Abcam Biochemicals, UK) were added to block N-methyl-D-aspartic acid (NMDA)

receptors, α-Amino-3-hydroxy-5-methyl-4-isoxazolepropionic acid (AMPA), and gamma aminobutyric acid A (GABAA) receptors, respectively.

## In vivo fiber photometry

Following AAV-EF1α-DIO-GCaMP6f injections, an optical fiber (125 μm outer diameter, 0.37 numerical aperture; Newdoon, Shanghai) was placed in a ceramic ferrule and was inserted toward the PVH. Fiber photometry (*Long et al., 2009*) uses the same fiber to both excite and record from GCaMP in real time. After surgery, mice were individually housed for at least 10 days to recover. Fluorescent signals were acquired with a laser beam passed through a 488 nm excitation laser (OBIS 488LS; Coherent), reflected off a dichroic mirror (MD498; Thorlabs), focused by an objective lens (Olympus), and coupled through a fiber collimation package (F240FC-A, Thorlabs) into a patch cable connected to the ferrule of an upright optic fiber implanted in the mouse via a ceramic sleeve (125 μm O.D.; Newdoon, Shanghai). GCaMP6 fluorescence was bandpass filtered (MF525–39, Thorlabs) and collected by a photomultiplier tube (R3896, Hamamatsu). An amplifier (C7319, Hamamatsu) was used to convert the photomultiplier-tube current output to voltage signals, which were further filtered through a low-pass filter (40 Hz cut-off; Brownlee 440). The photometry voltage traces were down-sampled using interpolation to match the EEG/EMG sampling rate of 512 Hz via a Power1401 digitizer and Spike2 software (CED, Cambridge, UK).

Photometry data were exported to MATLAB R2018b MAT files from Spike2 for further analysis. We segmented the value of the fluorescent change ($\Delta F/F$) by calculating $(F – F_0)/F_0$, where $F_0$ is the baseline of the fluorescent signal. We recorded data for 3–5 hr per mouse for the analysis of sleep–wake transitions to calculate the averaged calcium signal of $\Delta F/F$ during all times of vigilant states. For analyzing state transitions, we determined each sleep–wake transition and calculated $\Delta F/F$ in a ± 40 s window around that time point.

## Firing rate analysis

Electrophysiological data were filtered with a band-pass filter (300–6000 Hz) to obtain neuronal spikes. Single-unit activities were sorted according to a threshold and shape detector using principal component analysis via Offline Sorter software (Plexon Co, USA). The first two principal components of each spike on the two-dimensional plot of detected spike events were extracted. Waveforms with similar principal components were clustered via a K-means sorting method. The isolated cluster was considered as a single unit recorded from the same neuron. Spikes with inter-spike intervals < 2 ms were discarded. Cross-correlation histograms were used to eliminate cross-channel artifacts. NeuroExplorer software (version 5.0) was used to produce firing-rate rastergrams, and Prism (version 7.0) was used to produce firing-rate histograms.

## Autonomic function monitoring

The heart rate and core temperature of mice were measured by implantable telemetry devices (ETA-F10, Data Sciences International, USA), which allows long-time continuous electrocardiography (ECG) recordings in freely moving, awake mice. Surgical procedures were as follows: After administering chloral hydrate (360 mg/kg, i.p.) to all mice for anesthesia, we removed their body hair liberally from all intended incision sites and surgically scrubbed the incision sites with 75 % alcohol. The implant body portion of the device was positioned subcutaneously along the lateral flank between the forelimb and hind limb. The biopotential leads were subcutaneously tunneled from the abdominal incision to the desired ECG electrode locations. The negative lead and the positive lead were placed at the right pectoral muscle and the left caudal rib region, respectively. All skin incisions were closed using wound clips. After surgery, all mice were kept in a warm environment until the return of normal postures and behaviors.

Biopotential signal recordings started at 08:00, ended at 18:00, and were sampled every 20 s. Each mouse received either vehicle or CNO (3 mg/kg, C2041, LKT) treatment at 09:00 (inactive period) on the following two consecutive days. Recording results were manually reviewed and then analyzed using Dataquest ART Software (Data Sciences International, USA).

## Measurement of CRH and CORT levels by ELISA

After each mouse received either vehicle or CNO (3 mg/kg, C2041, LKT) treatment at 09:00 (inactive period), blood samples were collected at corresponding time points (2, 3, and 4 hr following injection)

and allowed to clot for two hours at room temperature before centrifugation for 15 min at 1000× g. Serum was removed and assayed immediately or aliquoted, and samples were stored at −80 °C. The concentration of CRH and CORT in serum was detected using enzyme-linked immunosorbent assay (ELISA) kits (CSB-E14068m and CSB-E07969m, CUSABIO Technology, China) according to the manufacturer's instruction.

## Histology and immunohistochemistry

To confirm the correct injection sites, after finishing all related experiments, each mouse was anesthetized with chloral hydrate (360 mg/kg, i.p.) and then perfused intracardially with 30 mL phosphate-buffered saline (PBS) followed by 30 mL 4 % paraformaldehyde (PFA). Their brains were removed and postfixed in 4 % PFA overnight and then incubated in 30 % sucrose phosphate buffer at 4 °C until they sank. Coronal sections (30 μm) were cut on a freezing microtome (CM1950, Leica, Germany), and the fluorescence of injection sites was checked with the location of the PVH according to the histology atlas of Paxinos and Franklin (2001, The Mouse Brain in Stereotaxic Coordinates 2nd edn [San Diego, CA: Academic]) using a microscope (Fluoview 1200, Olympus, Japan).

For dual immunostaining with c-fos and mCherry, mice were deeply anesthetized with chloral hydrate (360 mg/kg, i.p.) and were perfused with PBS followed by 4 % PFA in 0.1 M phosphate buffer. The brain was then dissected and fixed in 4 % PFA at 4 °C overnight. Fixed samples were sectioned into 30 μm coronal sections using a freezing microtome (CM1950, Leica, Germany). For immunohistochemistry, the floating sections were washed in PBS and were then incubated in the following primary antibodies in PBS containing 0.3 % Triton X-100 (PBST) at 4 °C: anti-rabbit c-fos (1:10,000 for 48 hr); primary antibody (Millipore); and anti-mouse NeuN (1:1000 for 12 hr; MAB377, Millipore). Primary antibodies were washed five times with PBS before incubation with secondary antibodies at room temperature for 2 hr (Alexa 488, 1:1000; Abcam). Finally, the sections were mounted on glass slides, dried, dehydrated, and cover-slipped. Fluorescent images were collected with a confocal microscope (Nikon AIR-MP).

## In situ hybridization

In situ hybridization with Vglut2 mRNA was performed via digoxigenin riboprobes in brain sections. The brain sections were placed onto slides and were surrounded by water-repellent traces. Subsequently, the slides were post-fixed in 4 % PFA for 20 min. In situ hybridization was processed as previously described (*Yuan et al., 2020*; *Zhang et al., 2019*). DNA templates for in situ hybridization probes were obtained by PCR from either wild-type-embryo or P0-mouse cDNA libraries. All the buffers contained 0.1 % of RNase inhibitor (diethyl pyrocarbonate, DEPC, B600154, Sangon Biotech). Finally, the sections were mounted on slides, dried, and coverslipped with VectaMount (Vector Laboratories). Images were acquired using a Leica confocal system or Olympus IX71 microscope. For DsRed immunohistochemistry combined with Vglut2 mRNA in situ hybridization, brain sections were placed onto slides and in situ hybridization was performed after DsRed immunohistochemistry was completed. All the buffers contained 0.1 % RNase inhibitor.

## Statistical analysis

All data are expressed as the mean ± standard error of the mean (SEM). Sample sizes were chosen based on previous studies (*Luo et al., 2018*; *Zhang et al., 2019*). Two-way repeated-measures analysis of variance (ANOVA) was used to perform group comparisons with multiple measurements. Paired and unpaired $t$ tests were used for single-value comparisons. One-way ANOVA was used to compare more than two groups, followed by post hoc Tukey tests for multiple pairwise comparisons. Prism 7.0 (GraphPad Software, San Diego, CA, USA) was used for all statistical analyses. $p < 0.05$ was considered statistically significant.

## Acknowledgements

We are grateful to Hui Dong, Ze-Ka Chen, Ya-Nan Zhao, Peng-Fei Xu, Ming-Jie Yang, Fan Yang, and An-Shu Chen for technical assistance.

## Additional information

### Funding

| Funder | Grant reference number | Author |
|---|---|---|
| National Key Research and Development Program of China | 2020YFC2005300 | Wei-Min Qu |
| National Natural Science Foundation of China | 82020108014 | Zhi-Li Huang |
| National Natural Science Foundation of China | 32070984 | Zhi-Li Huang |
| National Natural Science Foundation of China | 82071491 | Wei-Min Qu |
| National Natural Science Foundation of China | 31871072 | Wei-Min Qu |
| National Natural Science Foundation of China | 81671317 | Chang-Rui Chen |
| Shanghai Science and Technology Innovation Action Plan Laboratory Animal Research Project | 201409001800 | Zhi-Li Huang |
| National Natural Science Foundation of China | 31900738 | Lei Xiao |
| Shanghai Pujiang Program | 9PJ1401800 | Lei Xiao |
| Shanghai Municipal Science and Technology Major Project and ZJLab | 2018SHZDZX01 | Zhi-Li Huang |
| Program for Shanghai Outstanding Academic Leaders | | Zhi-Li Huang |

The funders had no role in study design, data collection and interpretation, or the decision to submit the work for publication.

### Author contributions

Chang-Rui Chen, Data curation, Formal analysis, Investigation, Methodology, Project administration, Software, Supervision, Visualization; Yu-Heng Zhong, Data curation, Formal analysis, Investigation, Methodology, Project administration, Visualization, Writing - original draft; Shan Jiang, Formal analysis, Investigation, Methodology, Project administration, Software, Validation, Visualization, Writing - original draft, Writing – review and editing; Wei Xu, Formal analysis, Investigation, Methodology, Project administration, Software; Lei Xiao, Investigation, Writing – review and editing; Zan Wang, Conceptualization, Methodology, Resources; Wei-Min Qu, Conceptualization, Data curation, Funding acquisition, Methodology, Resources, Supervision; Zhi-Li Huang, Conceptualization, Data curation, Funding acquisition, Investigation, Resources, Supervision, Validation, Writing – review and editing

### Author ORCIDs

Chang-Rui Chen (ID) http://orcid.org/0000-0001-9879-9620
Lei Xiao (ID) http://orcid.org/0000-0002-1640-9690
Zhi-Li Huang (ID) http://orcid.org/0000-0001-9359-1150

### Ethics

All animal experiments were approved by the Medical Experimental Animal Administrative Committee of Shanghai. All experimental procedures involving animals were approved by the Animal Experiment and Use Committee of Fudan University (20150119 - 067).

### Decision letter and Author response

Decision letter https://doi.org/10.7554/eLife.69909.sa1

Author response https://doi.org/10.7554/eLife.69909.sa2

## Additional files

### Supplementary files
• Transparent reporting form

### Data availability
All data generated or analyzed during this study are included in the manuscript and supporting files. Source data files have been provided at DRYAD (DOI: https://doi.org/10.5061/dryad.bg79cnpb6, https://doi.org/10.5061/dryad.x3ffbg7jw, https://doi.org/10.5061/dryad.r4xgxd2db).

The following dataset was generated:

| Author(s) | Year | Dataset title | Dataset URL | Database and Identifier |
|---|---|---|---|---|
| Huang Z, Chen C, Zhong Y, Jiang S, Xu W, Wang Z, Xiao L, Qu W | 2021 | EEG/EMG, Photometry and Electrophysiological data | https://doi.org/10.5061/dryad.bg79cnpb6 | Dryad Digital Repository, 10.5061/dryad.bg79cnpb6 |
| Huang Z, Chen C, Zhong Y, Jiang S, Xu W, Wang Z, Xiao L, Qu W | 2021 | electrophysiological data | https://doi.org/10.5061/dryad.x3ffbg7jw | Dryad Digital Repository, 10.5061/dryad.x3ffbg7jw |
| Huang Z, Chen C, Zhong Y, Jiang S, Xu W, Wang Z, Xiao L, Qu W | 2021 | CRH and CORT, heart rate, temperature | https://datadryad.org/stash/share/p6Zp9wqpJpy8yHqqPo9N46JGt3CuSx6n83zZMslv5f0 | Dryad Digital Repository, 10.5061/dryad.r4xgxd2db |

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
