## [Editor Report]

Your work demonstrating a novel role of a projection from the paraventricular nucleus of the hypothalamus to the parabrachial nucleus in regulating wakefulness will be of interest to the readership of *eLife*. In particular, your study will add to the fields of sleep and hypothalamic research.

---

## [Decision Letter]

**Decision letter after peer review:**

Thank you for submitting your article "Dysfunctions of the paraventricular hypothalamic nucleus induce hypersomnia in human and mice" for consideration by *eLife*. Your article has been reviewed by 3 peer reviewers, one of whom is a member of our Board of Reviewing Editors, and the evaluation has been overseen by Ma-Li Wong as the Senior Editor. The following individual involved in review of your submission has agreed to reveal their identity: Michael Lazarus (Reviewer #3).

Essential revisions:

1) The authors need to delineate the potential role of changes in endocrine and autonomic function underlie the change in wakefulness.

2) The inclusion of the human data needs to be further developed or justified. It may improve the manuscript by removing it.

3) More evidence for selective ablation of vglut2-positive neurons is needed.

4) More detail regarding the statistical analyses and power calculations are required.

5) The photomicrographs of the anterograde tracing are of poor quality and need to be improved.

*Reviewer #1:*

Chen and colleagues describe results from a series of studies in human subjects and mouse models investigating the role of the paraventricular nucleus of the hypothalamus in regulating sleep and wakefulness. They provide a series of studies that suggest that increased activity of the PVH increases wakefulness and that inhibition and/or ablation of the PVH induces hypersomnia. Overall, the studies are interesting and will be of interest. However, several issues need to be addressed.

The biggest criticism is that the human data though tantalizing is at this point preliminary. Obviously, finding patients with lesions centered in the hypothalamus are relatively rare. However, 3 patients is not adequately powered. The omission of the human data would strengthen the manuscript.

In all studies, how were changes in endocrine and autonomic function ruled out as the cause of the change in wakefulness? This is an important point as the PVH has projections to the median eminence and to autonomic preganglionic neurons in the brainstem and spinal cord. For example, do changes in glucocorticoid levels occur and does this help explain the long lasting effects of CNO?

Why were the optogenetic studies in the supplemental data (SFig 5).

The authors should address/discuss why single injection of CNO into mice expressing Dq DREADD in the PVH (Figure 4C and Fig5I) can affect sleep up to ~9 hours.

In the mouse studies, the methods appear to be solid. The used small volumes but it is unclear how injection sites were assessed in each case. Also, it is not clear when unilateral and bilateral injections were performed. More detail is required.

*Reviewer #2:*

Globally, the article is very interesting and show a huge number of convincing data.

The authors start by showing three patients with lesion in the PVH who did show hypersomnia. Although the analysis is very limited and won't be sufficient for a full demonstration, I found this small description of interest.

The authors then study the afferents to the PVH Glutamate neurons using specific retrograde tracing. They describe a very large number of inputs to these neurons.

The effect of inactivation of the PVH vglut2 neurons by chemogenetic is quite remarkable with a large decrease of Waking and an increase in NREM sleep without a REM increase. The excitation of the same neurons leads to the opposite effect, I.e. a long lasting induction of Waking and nearly disappearance of sleep. They also show that optogenetic activation of the Glu PVH neurons during NREM induced a rapid transition to Waking. They finally showed that optogenetic stimulation of the PVH Glu terminals in the LS and the PB also induced Waking. Chemogenetic stimulation of the CRH, Dynorphin or oxytocin neurons induced also an increase in Waking although with a much shorter duration than for the glutamate neurons. Then, the authors made an ablation of the glutamatergic neurons of the PVH using genetic tools. They found an increase of NREM and a decrease of Waking during the dark rather than during the light period. This is fitting with the patients data.

Using unit recordings and fiber photometry, they found out that PVH glutamatergic neurons are active during Waking, nearly inactive during NREM and show some activity during REM. The further then show the axonal projection from the PVH glutamatergic neurons using genetic tracing. This is welcome.

First a general comment, I’m missing in the Results section the number of animals and the statistics in the text. As it is, it is very descriptive and difficult to figure out the power of the results.

The neuroanatomical retrograde tracing analysis is very limited and in fact does not bring much to the demonstration. They conclude line 128-129 that the fact that the neurons receive inputs from the PVT, PB, ZI and VLPAG suggest that the PVH might act as key central node for sleep-wake regulation. I found this statement rather weak since so many structures project to the PVH that it is not that surprising that some could be interesting for waking. Further, the involvement in Wake of the mentioned structures is not that well demonstrated. I would prefer a more detailed description of the afferents and to couple it with cFos to determine whether they are wake active. It can be left in the paper as it is but it does not bring crucial information.

I wonder why they choose to study only two efferent structures and not others? Do they believe that only the projections to the given structures is inducing Waking?

I’m missing information on whether the three neuropeptides populations of neurons colocalize or not? This is not clear to me? It would be great to know whether each of them are subtypes of glutamatergic PVH neurons.

What is missing a bit in the unit recordings is a more detailed analysis on the discharge pattern during Waking. Indeed, it would be helpful to know whether the PVH Glu neurons change their activity during different behaviors to more deeply understand their function.

The photomicrographs shown for the anterograde tracing are of poor quality and it is difficult to figure out where we are in the brains. The drawings help but are not sufficient to show the location of the fibers. I would favor photomicrographs in which landmarks are visible. There is also no quantification there and it is missing. I wonder also whether there is a strong projection to the median eminence since these neurons are mainly known for their neuroendocrine function? I would like the authors to comment on such role of these neurons in the Discussion.

I never heard of the dorsal terminal nucleus. This does not seem the common name of this region. Please explain.

*Reviewer #3:*

The neuronal mechanisms of hypersomnolence disorder (HD), characterized by excessive sleep due to pathological conditions, remains essentially unknown. Based on human data from HD patients with lesions around the paraventricular hypothalamic nucleus (PVH), Chen et al. hypothesized that the PVH is a key node for sleep-wake regulation. In addition to the human data, the authors investigated the role of PVH glutamatergic neurons in the control of wakefulness in mice by using cutting-edge genetically engineered systems in combination with in-vivo recording of neuronal activity. They found that PVH-parabrachial nucleus and PVH-lateral septum projections mediated the wake-promoting effects of PVH glutamatergic neurons and ablation of the glutamatergic neurons leads to hypersomnia-like behaviors in mice. I feel that this is an important observation, and, due to combination of human and mouse data, the findings may have clinical relevance to treat hypersomnia. The experiments are well designed and the paper is in principle well written.

I have one major comment:

Evidence for selective ablation of vglut2-positive neurons (e.g. vglut2 in-situ hybridization) in figure 6B should be shown. NeuN staining is not enough.

---

## [Author Response]

Essential revisions:1) The authors need to delineate the potential role of changes in endocrine and autonomic function underlie the change in wakefulness.

As requested, we did more experiments and added a supplement figure. Please see Figure 3—figure supplement 2.

In order to delineate the potential changes in autonomic function, we measured the heart rate and temperature of PVH^vglut2-M3^ mice after treatment with vehicle or CNO by implantable telemetry devices. We found that chemogenetic activation of PVH^vglut2^ neurons induced an increase in heart rate and temperature for 3 h, which both peaked at about 2 h after injection and returned to baseline level at about 3 h following administration of CNO. As for the endocrine function, we detected serum corticotropin release factor (CRF) and corticosterone (CORT) levels in PVH^vglut2-M3^ mice after treatment with vehicle or CNO. As shown in Figure supplement 3C, chemogenetic activation of PVH^vglut2^ neurons significantly increased CRF level at 2 h, 3 h and 4 h timepoints after injection of CNO, while the CORT level was only higher than that of vehicle group at 3 h timepoint following administration.

Glucocorticoid hormones (CORT for mouse) are the effector hormones of the hypothalamic-pituitary-adrenal (HPA) axis neuroendocrine system, which are regulated by CRF (secreted in the medial parvocellular portion of the PVH) and produce direct negative feedback inhibition of CRF neurons in the PVH [1] (Robert LSpencer, et al., 2017, Figure 1A). Our results showed that although chemogenetic activation of PVH^vglut2^ neurons induced higher CRH levels, it did not cause a continuous high level of CORT. These results suggested that changes in glucocorticoid levels indeed occurred but did not last for 9 h, so the long-lasting effects of wakefulness induced by PVH glutaminergic neurons’ activation could only partly be explained by endocrine and autonomic function changes; the wake-promoting neural circuits of PVH^vglut2^ neurons played a more important role in the 9 h of wakefulness.

2) The inclusion of the human data needs to be further developed or justified. It may improve the manuscript by removing it.

We greatly appreciate your suggestion. We deleted the human data in the revised manuscript.

3) More evidence for selective ablation of vglut2-positive neurons is needed.

As requested, we supplemented evidence for selective ablation of vglut2-positive neurons by *in situ* hybridization. Please refer to the answer for Reviewer #3, Question #1.

4) More detail regarding the statistical analyses and power calculations are required.

We confirmed the number of animals and statistics in each experiment and revised figure legends according to the *eLife*’s format requirements.

5) The photomicrographs of the anterograde tracing are of poor quality and need to be improved.

As requested, we replaced high-resolution images of anterograde tracing in the revised manuscript. Please see Figure 1 in the revised manuscript.

Reviewer #1:The authors should address/discuss why single injection of CNO into mice expressing Dq DREADD in the PVH (Figure 4C and Fig5I) can affect sleep up to ~9 hours.

Thank you for your suggestion. First, given that PVH glutaminergic neurons are largely co-expressed with corticotropin-releasing hormone (CRH), prodynorphin (PDYN) and oxytocin (OT) neurons, which are all wake-promoting neurons, the 9-h of wakefulness induced by activation of PVH^vglut2^ neurons partly results from activation of these types of neurons. Furthermore, the PVH sends direct projections to the PB, and it was reported that chemogenetic activation of the PB produced ~11 h of continuous wakefulness in rats [2, 3]. Furthermore, as we mentioned in response to Essential Revisions 1, the change of endocrine and autonomic function may also contribute to the long-time wakefulness. All these have been added to the discussion in revised manuscript. Please see pages 11-12, lines 286-302.

In the mouse studies, the methods appear to be solid. The used small volumes but it is unclear how injection sites were assessed in each case. Also, it is not clear when unilateral and bilateral injections were performed. More detail is required.

Thank you for your suggestion. According to our previous studies [4-7], after finishing all related experiments, each mouse was anesthetized with chloral hydrate (360 mg/kg) and then perfused intracardially with 30 mL phosphate-buffered saline (PBS) followed by 30 mL 4% paraformaldehyde (PFA). Their brains were removed and postfixed in 4% PFA overnight and then incubated in 30% sucrose phosphate buffer at 4°C until they sank. Coronal sections (30 μm) were cut on a freezing microtome (CM1950, Leica, Germany), and the fluorescence of injection sites was checked with the location of the PVH according to the histology atlas of Paxinos and Franklin (2001, The Mouse Brain in Stereotaxic Coordinates 2nd edn [San Diego, CA: Academic]) using a microscope (Fluoview 1200, Olympus, Japan). Method details were added in the revised manuscript. Please see page 19, lines 485-493.

Unilateral injections were performed in in vivo fiber photometry (AAV-DIO-GCamp6f), anterograde tracing (AAV-DIO-eGFP), while bilateral injections were used in chemogenetic and optogenetic manipulation (AAV-DIO-ChR2, AAV-DIO-hM3Dq, AAV-DIO-hM4Di) and ablation experiments (AAV-DIO-Caspase3). We added these in the revised manuscript. Please see page 14, lines 350-353.

Reviewer #2:First a general comment, I'm missing in the Results section the number of animals and the statistics in the text. As it is, it is very descriptive and difficult to figure out the power of the results.

Thank you for your comments. We added the number of animals and statistics in each experiment and revised figure legends according to the *eLife*’s format requirements.

The neuroanatomical retrograde tracing analysis is very limited and in fact does not bring much to the demonstration. They conclude line 128-129 that the fact that the neurons receive inputs from the PVT, PB, ZI and VLPAG suggest that the PVH might act as key central node for sleep-wake regulation. I found this statement rather weak since so many structures project to the PVH that it is not that surprising that some could be interesting for waking. Further, the involvement in Wake of the mentioned structures is not that well demonstrated. I would prefer a more detailed description of the afferents and to couple it with cFos to determine whether they are wake active. It can be left in the paper as it is but it does not bring crucial information.

Thank you for your suggestions. We found that PVH^vglut2^ neurons received direct inputs from many brain regions (Figure 1E and Figure supplement 1). Among them, the PVT expressed a higher level of c-fos during the active period (24:00) than the inactive period (12:00), and activation of PVT^vglut2^ neurons induced rapid transitions from sleep to wakefulness [8]. Similarly, chemogenetic activation of PB neurons induced increased expression of c-fos and a potent wakefulness lasting about 9 hours [2]. In contrast, ablation of PB^vglut2^ neurons induced hypersomnia-like behaviors [9].

More importantly, we observed higher c-fos expression in the PVH during the active period (23:00) than during the inactive period (11:00) (Figure 1A, B). Therefore, we speculated that PVH^vglut2^ neurons might act as a central node for sleep–wake regulation.

I wonder why they choose to study only two efferent structures and not others? Do they believe that only the projections to the given structures is inducing Waking?

Thank you for your insightful questions. The reason for choosing these two efferent structures is that PVH^vglut2^ neurons mainly projected to four neuroanatomical sites related to sleep–wake regulation: the PB, LSv, PVT and NTS (Figure 5—figure supplement 1C and Figure 5—table supplement 1). The PVT, PB and NTS have been demonstrated to be essential in controlling wakefulness [2, 3, 8] and the PB innervates PVH^vglut2^ neurons to form bidirectional connections. In addition, the PVH^vglut2^**→**LSv pathway is involved in the regulation of feeding, which is a wakefulness-required behavior[10]. Therefore, we explored these four pathways to identify the neuronal circuits mediating the wake-promoting effect of PVH^vglut2^ neurons and found that optical stimulation of PVH^vglut2^**→**PB and PVH^vglut2^**→**LSv promoted wakefulness (Figure 5) while activation of PVH^vglut2^**→**NTS or PVH^vglut2^**→**PVT pathways did not alter sleep–wake states (Figure 5—figure supplementary 2).

I'm missing information on whether the three neuropeptides populations of neurons colocalize or not? This is not clear to me? It would be great to know whether each of them are subtypes of glutamatergic PVH neurons.

Thank you for your essential suggestions. The distributions of neuropeptides populations in the PVH have been reported in previous research [11], in which Xu et al., observed marker gene enrichment and pairwise gene co-expression in the PVH regions (Figure S5, Xu et al., 2020). According to the results, almost 100% of neurons expressing CRH, PDYN and OXT co-expressed with Vglut2 in the middle PVH (mPVH) and posterior PVH (pPVH), which were also the main distribution regions of these three neuropeptides populations in the PVH (Figure 2B, Xu et al., 2020).

What is missing a bit in the unit recordings is a more detailed analysis on the discharge pattern during Waking. Indeed, it would be helpful to know whether the PVH Glu neurons change their activity during different behaviors to more deeply understand their function.

Thank you for your insightful questions. In the near future, we would like to analyze them to explore other function of PVH^vglut2^ neurons.

The photomicrographs shown for the anterograde tracing are of poor quality and it is difficult to figure out where we are in the brains. The drawings help but are not sufficient to show the location of the fibers. I would favor photomicrographs in which landmarks are visible. There is also no quantification there and it is missing. I wonder also whether there is a strong projection to the median eminence since these neurons are mainly known for their neuroendocrine function? I would like the authors to comment on such role of these neurons in the Discussion.

Thank you for your insightful questions. We have supplemented high-resolution images of anterograde tracing in the revised manuscript. As requested, we provided complementary quantification (“+” represents the fluorescence intensity of axon terminal from PVH^vglut2^ neurons). Additionally, we found that PVH^vglut2^ neurons project to the median eminence, please see Figure 5—figure supplement 1C and Figure 5—table supplement 1. Related discussion has been added in the revised manuscript (pages 12, lines 303-306).

I never heard of the dorsal terminal nucleus. This does not seem the common name of this region. Please explain.

Sorry for the mistake, it should be “nucleus of the solitary tract”. We corrected all related statements in the revised manuscript.

Reviewer #3:I have one major comment:Evidence for selective ablation of vglut2-positive neurons (e.g. vglut2 in-situ hybridization) in figure 6B should be shown. NeuN staining is not enough.

We greatly appreciate your suggestion. As requested, we supplemented evidence for selective ablation of vglut2-positive neurons by *in-situ* hybridization. Please refer to the Figure 7E.

References

1. Spencer RL, Deak T. A users guide to HPA axis research. Physiol Behav. 2017;178:43-65.

2. Qiu MH, Chen MC, Fuller PM, Lu J. Stimulation of the Pontine Parabrachial Nucleus Promotes Wakefulness via Extra-thalamic Forebrain Circuit Nodes. Curr Biol. 2016;26(17):2301-12.

3. Xu Q, Wang DR, Dong H, Chen L, Lu J, Lazarus M, et al. Medial Parabrachial Nucleus Is Essential in Controlling Wakefulness in Rats. Front Neurosci. 2021;15:645877.

4. Li YD, Luo YJ, Xu W, Ge J, Cherasse Y, Wang YQ, et al. Ventral pallidal GABAergic neurons control wakefulness associated with motivation through the ventral tegmental pathway. Mol Psychiatry. 2020.

5. Luo YJ, Li YD, Wang L, Yang SR, Yuan XS, Wang J, et al. Nucleus accumbens controls wakefulness by a subpopulation of neurons expressing dopamine D(1) receptors. Nat Commun. 2018;9(1):1576.

6. Zhong YH, Jiang S, Qu WM, Zhang W, Huang ZL, Chen CR. Saikosaponin a promotes sleep by decreasing neuronal activities in the lateral hypothalamus. J Sleep Res. 2021:e13484.

7. Bao WW, Xu W, Pan GJ, Wang TX, Han Y, Qu WM, et al. Nucleus accumbens neurons expressing dopamine D1 receptors modulate states of consciousness in sevoflurane anesthesia. Curr Biol. 2021;31(9):1893-902.e5.

8. Ren S, Wang Y, Yue F, Cheng X, Dang R, Qiao Q, et al. The paraventricular thalamus is a critical thalamic area for wakefulness. Science. 2018;362(6413):429-34.

9. Kaur S, Pedersen NP, Yokota S, Hur EE, Fuller PM, Lazarus M, et al. Glutamatergic signaling from the parabrachial nucleus plays a critical role in hypercapnic arousal. J Neurosci. 2013;33(18):7627-40.

10. Xu Y, Lu Y, Cassidy RM, Mangieri LR, Zhu C, Huang X, et al. Identification of a neurocircuit underlying regulation of feeding by stress-related emotional responses. Nat Commun. 2019;10(1):3446.

11. Xu S, Yang H, Menon V, Lemire AL, Wang L, Henry FE, et al. Behavioral state coding by molecularly defined paraventricular hypothalamic cell type ensembles. Science. 2020;370(6514).